



# The improvements to the regional South China Sea Operational Oceanography Forecasting System (SCSOFSv1)

Xueming Zhu[1 & 2], Ziqing Zu[1], Shihe Ren[1*], Miaoyin Zhang[1], Yunfei Zhang[1], Hui Wang[3,1*], Ang Li[1]

[1]National Marine Environmental Forecasting Center, Key Laboratory of Marine Hazards Forecasting, Ministry of Natural Resources, Beijing, 100081, China

[2]Southern Marine Science and Engineering Guangdong Laboratory (Zhuhai), Zhuhai, 519000, China

[3]Institute of Marine Science and Technology, Shandong University, Qingdao, Shandong, 266237, China

*Correspondence to*: Shihe Ren (rensh@nmefc.cn), Hui Wang(wangh@nmefc.cn)

**Abstract.** South China Sea Operational Oceanography Forecasting System (SCSOFS) had been constructed and operated in National Marine Environmental Forecasting Center of China to provide daily updated hydrodynamic forecasting in SCS for the future 5 days since 2013. This paper presents recent comprehensive updates of the configurations of the physical model and data assimilation scheme in order to improve SCSOFS forecasting skills. It highlights three of the most sensitive updates, including sea surface atmospheric forcing method, tracers advection discrete scheme, and modification of data assimilation scheme. Inter-comparison and accuracy assessment among five versions during the whole upgrading processes are performed by employing OceanPredict Inter-comparison and Validation Task Team Class4 metrics. The results indicate that remarkable improvements have been achieved in SCSOFSv2 with respect to the original version known as SCSOFSv1. Domain averaged monthly mean root mean square errors decrease from 1.21 ℃ to 0.52 ℃ for sea surface temperature, from 21.6cm to 8.5cm for sea level anomaly, respectively.

## 1. Introduction

The South China Sea (SCS) is located between 2°30′S～23°30′N and 99°10′E～121°50′E, the largest in area and the deepest in depth, a semi-closed marginal sea in the western Pacific. Its area is about 3.5 million km², and its maximum depth is about 5300 m at the central region. It connects to the East China

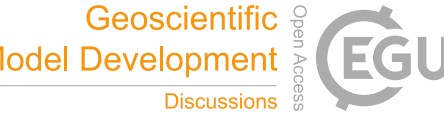

Sea by the Taiwan Strait to the northeast, to the North Pacific Ocean by the Luzon Strait (LUS) to the

east, to the Java Sea by the Karimata Strait to the south. Numerous islands, irregular and complex coastal

boundaries, and drastic changes in bottom topography all together contribute to the great complex

distribution of topography in the SCS.

The upper-layer basin-scale ocean circulations of the SCS are mainly controlled by the East Asian

Monsoon (Hellerman and Rosenstein, 1983), showing a cyclonic gyre in winter and an anti-cyclonic gyre

in summer (Mao et al., 1999;Chu and Li, 2000). The multi-scale oceanic circulation dynamical processes

of the SCS are affected by various factors, i.e. the Kuroshio intrusion through the LUS (Nan et al.,

2015;Farris and Wimbush, 1996;Liu et al., 2019), internal waves (Li et al., 2011;Li et al., 2015) or

internal solitary waves (Zhang et al., 2018;Zhao and Alford, 2006;Cai et al., 2014) generated in the LUS

and propagating in the northern SCS, the SCS throughflow as a branch of the Pacific to Indian Ocean

throughflow (Wei et al., 2019;Wang et al., 2011), and energetic mesoscale eddy activities (Zu et al.,

2019;Xu et al., 2019;Zhang et al., 2016;Zheng et al., 2017;Hwang and Chen, 2000;Wang et al., 2020).

The multi-scale dynamical mechanisms in the SCS are too complex to understand clearly as yet, it has

always been a challenge to simulate or reproduce the ocean circulations, not to mention forecast future

oceanic status by Operational Oceanography Forecasting System (OOFS).

Within coordination and leadership of Global Ocean Data Assimilation Experiment (GODAE)

OceanView (GOV, https://www.godae-oceanview.org; Tonani et al., 2015;Dombrowsky et al., 2009), in

recent decade or two, several regional OOFSs have been developed and operated based on the state-of-

the-art community numerical ocean models in different regions of the ocean. Tonani et al. (2015)

summarized that there were 19 regional systems running operationally in total till 2015.

For instance, Canadian Operational Network of Coupled Environmental Prediction Systems

(CONCEPTS) from Canada was built based on the Nucleus for European Modelling of the Ocean

(NEMO) 3.1, whose domain covered the Arctic and North Atlantic with 1/12° horizontal resolution; the

Real-Time Ocean Forecast System (RTOFS) from US National Oceanic and Atmospheric

Administration (NOAA) National Centers for Environmental Prediction (NCEP) was designed based on

the HYbrid Coordinate Ocean Model (HYCOM) and implemented in the North Atlantic on a curvilinear





coordinate, with the resolution ranging from 4 km to 18 km in horizontal; The Meteorological Research

Institute (MRI) of Japan Meteorological Agency (JMA) developed the Multivariate Ocean Variational

Estimation System/MRI Community Ocean Model (MOVE/MRI.COM) coastal monitoring and

forecasting system based on the MRI.COM (Tsujino et al., 2006). The model consists of a fine-resolution

(2km) coastal model around Japan and an eddy-resolving (10km) Western North Pacific (WNP) model

with one-way nesting; the Chinese Global operational Oceanography Forecasting System (CGOFS) was

developed and operated based on the Regional Ocean Modelling System (ROMS, Shchepetkin and

McWilliams, 2005) and NEMO by National Marine Environmental Forecasting Center, covering 6

subdomains from global to polar regions, Indian Ocean, Northwest Pacific, Yellow Sea and East China

Sea (Kourafalou et al., 2015), South China Sea (Zhu et al., 2016), with their horizontal resolutions

ranging from 1/12° to 1/30°. It is worth noting that there are considerable differences among those

systems in many aspects, such as the model codes, area coverage, horizontal/vertical resolutions, data

assimilation schemes, and so on, according to the user needs or regional ocean characteristics.

In order to better satisfy end users' needs, OOFSs need to be updated and improved continuously since

operation. In general, most improvements of OOFSs are implemented by increasing horizontal or vertical

grid resolution, changing the data assimilation schemes from a simple one to sophisticate one,

assimilating more amounts and types of observation data, by benefiting from the growth of high-

performance computing power and global or regional observation network. Initially, the

MOVE/MRI.COM was developed based on a three-dimensional variational (3DVAR) analysis scheme

and implemented in 2008 (Usui et al., 2006), then it is updated to the four-dimensional variational

(4DVAR) analysis scheme to provide better representation of mesoscale processes (Usui et al., 2017).

Mercator Ocean International (MOI) global monitoring and forecasting system had been routinely

operated in real time with an intermediate-resolution at 1/4° and 50 vertical levels since early 2001. An

upgrading of increasing horizontal resolution was implemented in December 2010, to consist a 1/12°

nested model over the Atlantic and Mediterranean. Real time daily services with a global 1/12° high-

resolution eddy-resolving analysis and forecasting were delivered by an updated system, since 19

October, 2016. Moreover, MOI also continues to implement regularly update by increasing system's



complexity, such as expanding the geographical coverage, improving models and assimilating schemes, and have developed several versions for the various milestones of the MyOcean project and the Copernicus Marine Environment Monitoring Service (CMEMS, Lellouche et al., 2013;Lellouche et al., 2018).

As mentioned in the literature of Zhu et al. (2016), the regional SCS Operational Oceanography Forecasting System (SCSOFS, here after named it as SCSOFSv1) has been developed and routinely operated in real time since the beginning of 2013. It has continued to be upgraded by modifying model settings in many aspects, such as mesh distributions, surface atmospheric field forcing, open boundary inputs, and so on, and improving data assimilation scheme according to the results of comparing and

validating from Zhu et al. (2016), in order to provide better services. The primary purpose of this paper is to introducing updates applied to SCSOFS, but only show the highest impact on the system. The other results from routine system updates or improvements will not be illustrated or discussed in detail.

This paper is organized as follows. A detailed description of some general/basic updates applied to SCSOFS will be provided in Section 2. Some highlights and sensitive updates and their impacts to the

performance of system are shown in Section 3. Results of the inter-comparison and assessment for different SCSOFS versions during the upgrading processes based on the 'Class 4 metrics' verification framework (Hernandez et al., 2009) will be shown in Section 4. Section 5 contains a summary of the scientific improvements and future plans for the next step.

## 2. Physical model description, updates and datasets

This section describes some general updates applied to the SCSOFSv1 in recent couple years. The newly updated system is named as SCSOFSv2 here after. In order to isolate the contribution of one modification, different simulations were performed for respective updates. However, some updates have been implemented directly according to model experiences or theory knowledges, without standalone evaluation. The performances from a few integrated updates will be shown in Section 4 for different

upgrading stages.





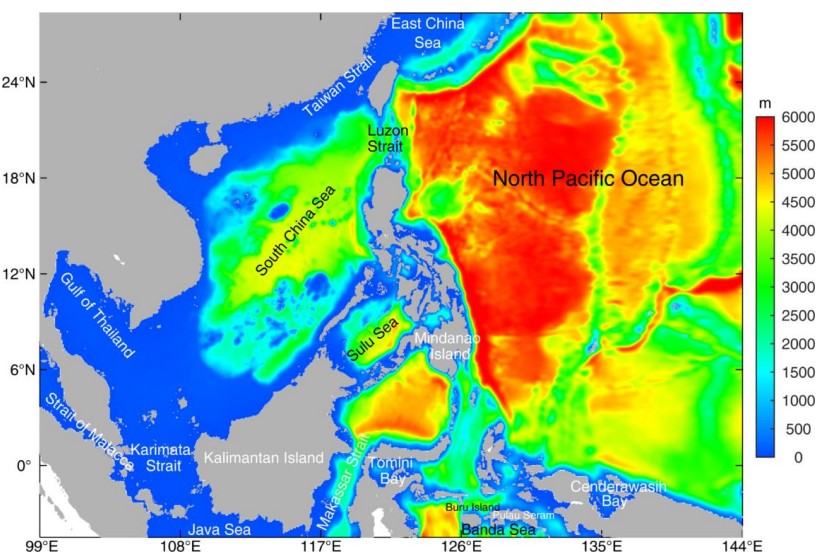

**Figure 1: The model domain and bathymetry of SCSOFSv2**

The SCSOFSv2 is still built based on ROMS, while whose version has been updated from v3.5 (svn trunk revision 648 in 2013) to v3.7 (svn trunk revision 874 in 2017). ROMS v3.7 incorporates some

changes for the model settings, which facilitating the operational running especially, besides of the major overhaul of the nonlinear, tangent linear, representor, multiple-grid nesting and adjoint numerical kernels. Firstly, we redistributed the land-sea grid mask layout to enable systems mesh land boundary fit the actual coastline better (Fig.1). By comparing with the Fig. 1 from Zhu et al. (2016), a few sea areas had been changed from land to sea or inverse, e.g. along the coast of China mainland, the Vietnam and the

Gulf of Thailand, around the coast of the Kalimantan Island and the Mindanao Island. In addition, the Strait of Malacca had been opened to connect with the Karimata Strait, and the western lateral boundary was treated as open boundary across the Strait of Malacca along 99°E, instead of closed boundary as in SCSOFSv1; along the south lateral open boundary, the Java Sea was connected to the Makassar Strait in the southeast of the Kalimantan Island, the Banda Sea was connected across the south of Buru Island and

Pulau Seram; and involved the Tomini Bay and the Cenderawasih Bay. It is obvious that the land-sea masks changing can generate significant effects on the sea water volume transportation in the model domain, thus would contribute to better simulation of ocean circulations.

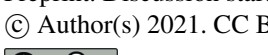



The bathymetry is replaced by the General Bathymetric Chart of the Oceans (GEBCO_2014 Grid) global continuous terrain model for ocean and land, which is with 30 arc-second spatial resolution in SCSOFSv2,

from ETOPO1 data set in SCSOFSv1, which is with 1 arc-minute grid resolution from U.S. National Geophysical Data Center (NGDC) . It is also merged with the measured topographic data in the coastal areas along China mainland, and adjusted with the tidal range. Then it is smoothed by applying a selective filter 8 times to reduce the isolated seamounts on the deep ocean, so that the "slope parameter" $r=\Delta h/2h$ is lower than a maximum value $r_0=0.2$ for each grid (Beckmann and Haidvogel, 1993; Marchesiello et

al., 2009), in order to supress the computational errors of the pressure-gradient (Shchepetkin and McWilliams, 2003). Then the two grid stiffness ratios parameters, slope parameter ($r$) and Haney number, change from 0.22 and 9.78 in SCSOFSv1 to 0.17 and 13.80 in SCSOFSv2, respectively. The maximum depth is set to be 6000m still, but the minimum depth changed from 10m in SCSOFSv1 to 5m in SCSOFSv2 (Wang, 1996). The final smoothed bathymetry is shown in Fig.1.

For the vertical terrain-following coordinate, it has been increased from 36 s-coordinate layers in SCSOFSv1 to 50 layers in SCSOFSv2. The transformation equation from the original formulation is also changed to an improved solution (Shchepetkin and McWilliams, 2005). The original vertical stretching function (Song and Haidvogel, 1994) is replaced with an improved double stretching function (Shchepetkin and McWilliams, 2005), to make it preserve a sufficient resolution in the upper 300m in

order to resolve the thermocline well. In this case, the thinnest layer changes from 0.16m in SCSOFSv1 to 0.09m in SCSOFSv2 near the surface.

The new initial temperature and salinity (T/S) fields in SCSOFSv2 are extracted from the Generalized Digital Environmental Model Version 3.0 (GDEMV3, Carnes, 2009) global climatology monthly mean in January, to substitute the version 2.2.4 of Simple Ocean Data Assimilation (SODA, Carton and Giese,

2008) datasets. All four lateral boundaries are open, whose temperature, salinity, velocity and elevation are prescribed by spatial interpolation from the new SODA 3.3.1 for the running 2005-2015 and SODA 3.3.2 for the running 2016-2018 datasets (Carton et al., 2018), instead of the original SODA 2.2.4. In this present, we use the SODA 3.3.1/2 monthly mean ocean state variables, which are mapped onto the regular



1/2°×1/2° Mercator horizontal grid from the original approximately 1/4°×1/4° displaced pole non-Mercator horizontal grid at 50 z vertical levels.

For the surface atmospheric forcing, we replace the dataset from the NCEP Reanalysis 2 provided by the NOAA/OAR/ESRL PSL, Boulder, Colorado, USA, accessible from their website at https://psl.noaa.gov/ (Kanamitsu et al., 2002), with 6-hourly Climate Forecast System Reanalysis (CFSR, Saha et al., 2010) for 2005-2011 and Climate Forecast System Version 2 (CFSv2, Saha et al., 2014) for 2011-2018. Both are archived at the National Center for Atmospheric Research (NCAR), Computational and Information Systems Laboratory, Boulder, Colorado, with a 0.2°-0.3° significantly higher horizontal grid than the 2.5°×2.5° resolution for NCEP Reanalysis 2.

The net surface heat flux correction is still following Barnier et al. (1995)'s method in SCSOFSv2, but the parameter (dQ/dSST) of kinematic surface net heat flux sensitivity to sea surface temperature (SST) is calculated using SST, sea surface atmospheric temperature, atmospheric density, wind speed and sea level specific humidity, instead of setting a constant number of -30 W m$^2$ K$^{-1}$ for the whole domain as in SCSOFSv1. So the parameter dQ/dSST varies temporally and spatially. Meanwhile, we use the infrared Advanced Very High Resolution Radiometer (AVHRR) satellite data in SCSOFSv2, which is an analysis constructed by combining observations from different platforms on a regular grid via optimum interpolation and provided by National Centers for Environmental Information (NCEI), instead of using the merged satellite's infrared sensors and microwave sensor, and *in-situ* (buoy and ship) data global daily SST (MGDSST) obtained from the Office of Marine Prediction of the Japan Meteorological Agency (JMA) in SCSOFSv1.

The North Equatorial Current (NEC) is an interior Sverdrup steady current in the subtropical NP and located at about 10°N-20°N, and usually bifurcates into two branches after encountering the western boundary along the Philippine coast in the west of 130°E (Qiu and Chen, 2010). However, the NEC separated into two branches in SCSOFSv1 affected by model eastern lateral boundary setting, its main branch located at about 9.5°N-13°N, the other branch located at 14.5°N-17°N (Fig. 2a), which is clearly not in line with the fact. The cause for above result is that the Guam Island (shown in red circle in Fig. 2, located about 13°26′N, 144°43′E) is included in SCSOFSv1, whose location is too close to the eastern



lateral boundary. To resolve this problem, the eastern lateral boundary has been moved westward from 145°E to 144°E to narrow the model domain and exclude the Guam Island in SCSOFSv2. It is found that the simulated NEC keeps the form of one main current until 130°E, then bifurcates into the southward-flowing Mindanao Current and the northward-flowing Kuroshio in SCSOFSv2 (Fig. 2b). Also, it is

shown that the Kuroshio of eastern Philippine Island and ocean circulations of northeastern SCS get stronger while the island of Guam removed. It indicates that the location of lateral open boundary is very important to the results of model's simulation and would better be set to far away from island enough, especially while the island located in the major ocean circulations.

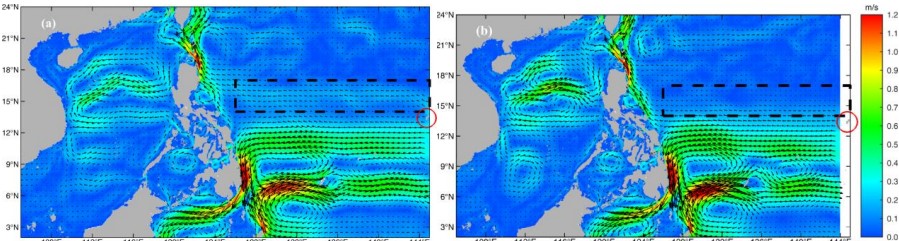

**Figure 2: The multi-year monthly mean sea surface currents (color shaded for current speed (m s-1), arrows for current direction) with vertical averaged above 100m in May. The left panel (a) is from SCSOFSv1 with the model domain including the Guam Island, the right panel (b) is from SCSOFSv1.2 with the eastern lateral boundary moving 1 deg westward.**

For the advection schemes of momentum, third-order upstream and fourth-order centered schemes are

used in both horizontal and vertical, respectively. Harmonic mixing scheme is used for both viscosity for momentum and diffusion for tracers in horizontal. Mellor-Yamada Level-2.5 vertical turbulent mixing closure scheme is used for both momentum and tracers. All of them in SCSOFSv2 are set to be same as in SCSOFSv1.

The SCSOFSv2 is run with 5s time step for the external mode, and 150s for the internal mode under all

new configurations mentioned above and in Section 3. The reason for modifying time step is related to the change of the discrete schemes, which will be illustrated in Section 3. A 26 years climatology run is conducted for spinning-up at first, and followed by a hindcast run from 2005 to 2018 (Wang et al., 2012). The daily mean of model results is archived and used for subsequent evaluation.





### 3.    Highlights and sensitive updates and their impacts

Most of bias or errors in the operational systems are mainly induced by some major recurring problems, for example sea surface atmospheric forcing, intrinsic deficiencies of numerical model (e.g., discrete schemes, parameterization schemes for sub-grid scale), initial errors, and the assimilation schemes. In this section, we elaborate solutions, that are not mentioned in Sect.2, to such problems applied in SCSOFSv2.

**3.1 Sea surface atmospheric forcing**

The air-sea interaction is one of the most essential physical processes that affect vertical mixing and thermal structure of the upper-ocean. The air-sea fluxes mainly include momentum flux, fresh water flux and heat flux. SST is an important indicator of ocean circulation, ocean front, upwelling and sea water mixing, whose variation mainly depending on the air-sea interaction, the ocean thermal and dynamical

factors (Bao et al., 2002). Thus, for OOFS and ocean numerical modelling, simulation and forecast accuracy of SST is one important metric to evaluate the modelling and forecasting performance.

The accurate input of sea surface atmospheric forcing plays a key role to excel in model simulation of SST. ROMS provides two methods to introduce sea surface atmospheric forcing: one is directly forcing ocean model by providing momentum fluxes (wind stress), net fresh water fluxes, net heat fluxes and

shortwave radiation fluxes from atmospheric datasets; the other is employing the COARE3.0 bulk algorithm (Fairall et al., 2003) to calculate air-sea momentum, fresh water and heat turbulent fluxes using the set of atmospheric variables from atmospheric datasets including wind speed at 10m above sea surface, mean sea level air pressure, air temperature at 2m above sea surface, air relative humidity at 2m above sea surface, downward longwave radiation flux, precipitation rate and shortwave radiation fluxes

(Large and Yeager, 2009). Since the SST using in the calculation of those three air-sea fluxes is extracted from ocean model, the increase of SST induces the variations of sensible heat flux, latent heat flux, and longwave radiation as a result, which then lead to increasing loss of ocean heat, and inhibiting further increase of SST, and vice versa. It means that an effective negative feedback mechanism could form between SST and SST-related heat fluxes. In this case, the simulated SST could maintain at a reasonable



level. The first method is employed in SCSOFSv1, and the second, bulk algorithm, is employed in

SCSOFSv2.

In order to evaluate the performances of different sea surface atmospheric forcing methods, we conduct

a special experiment by changing the method based on SCSOFSv1, here named the experiment as

BulkFormula. In this experiment, we use the merged satellite SST analysis with a multi-scale optimal

interpolation called the Operational SST and Sea Ice Analysis (OSTIA) system, which globally coverage

on a daily basis at a horizontal grid resolution of 1/20° (~6 km) and provided by the Met Office (Donlon

et al., 2012), to verify the results of SCSOFS.

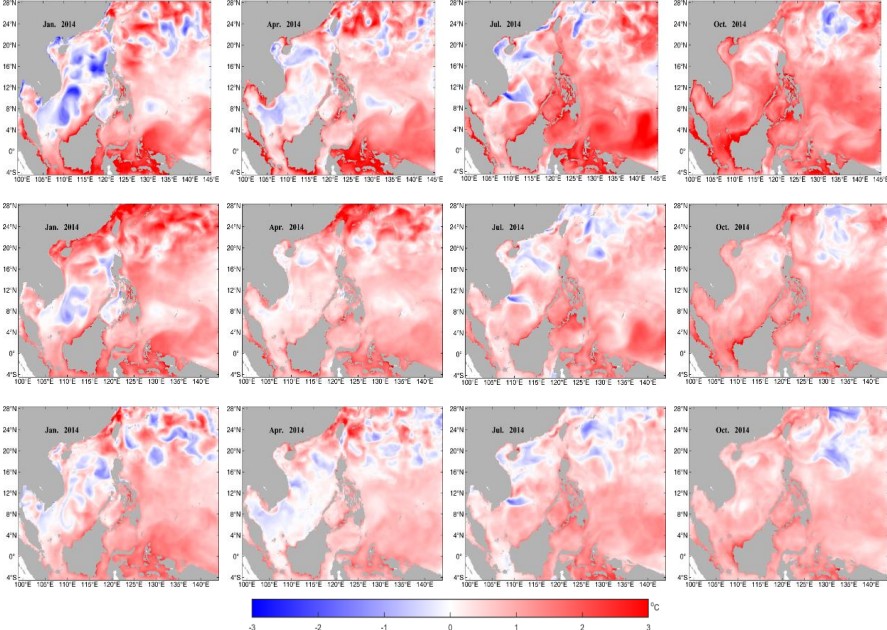

**Figure 3: The monthly mean SST differences in January, April, July, and October, 2014: SCSOFSv1 minus**
**OSTIA SST (upper panels), BulkFormula minus OSTIA SST (middle panels), SCSOFSv2 minus OSTIA SST**
**(lower panels)**

Figure3 shows the distributions of monthly mean SST differences in January, April, July and October,

2014 to stand for Winter, Spring, Summer and Autumn, respectively. SST differences are calculated with

SCSOFSv1, BulkFormula, and SCSOFSv2 subtracts OSTIA SST, respectively. It is found that the

simulated SST are higher than OSTIA SST for all three sets of results. The difference from SCSOFSv1





is pronouncedly higher than the differences from BulkFormula and SCSOFSv2. The maximum differences mainly occur near coast (Fig.3 upper panels), especially for a few bays embedded into the mainland which is hard to be resolved well with 2-3 horizontal grids at 1/30° resolution and in very shallow water depth in SCSOFSv1. This is because sea surface atmospheric forcing data is not accurate enough near the coast, and provide abnormally more heat to ocean causing the continuously heating up of coastal water. Thus, simulated SST is beyond normal level in SCSOFSv1. This phenomenon can be alleviated significantly by introducing the effective negative feedback mechanism between model's SST and air-sea heat flux by employing the COARE 3.0 bulk algorithm, which is employed in both BulkFormula and SCSOFSv2 (Fig.3 middle and lower panels).


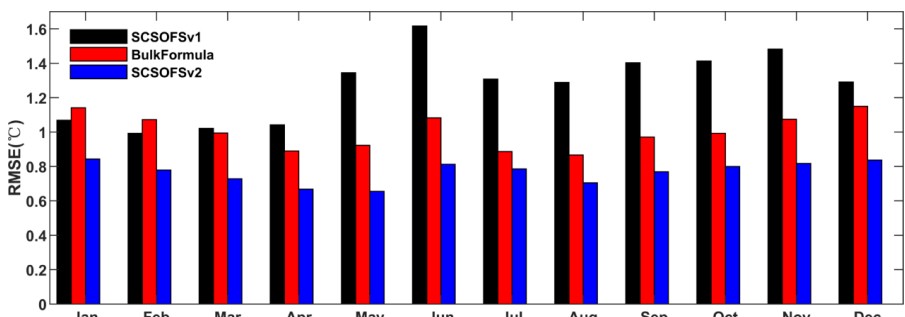

**Figure 4: Domain averaged monthly mean SST RMSE comparison among SCSOFSv1(black), BulkFormula (red), SCSOFSv2 (blue) and OSTIA SST in January, April, July, and October, 2014**


Figure 4 shows bars of domain averaged Root-Mean-Square Error (RMSE) of monthly mean SST differences of SCSOFSv1, BulkFormula, SCSOFSv2 with respect to OSTIA datasets in each month of 2014. It is found that the domain averaged RMSE of monthly mean SST differences from SCSOFSv1 is about 0.99-1.62°C, the annual mean value is about 1.27°C. The highest (1.62°C) is in June, the lowest (0.99°C) is in February. Monthly mean RMSE for BulkFormula run is about 0.87-1.15°C, the annual mean value is about 1.00°C, the maximum value (1.15°C) is in January and December, the minimum value (0.87°C) is in August. The performance of model skill for the annual mean SST RMSE can be improved by about 21% only by changing the method of sea surface atmospheric forcing from directly forcing to COARE 3.0 bulk algorithm due to effective negative feedback mechanism.



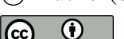



However, domain averaged RMSE of monthly mean SST differences from SCSOFSv1 is lower than that

from BulkFormula in January and February. It indicates that COARE 3.0 bulk algorithm does not work

better than direct forcing at all time, even with effective negative feedback mechanism. This may be

surface forcing field data dependent, and the accurate dataset of sea surface atmospheric forcing is more

effective than the forcing methodology selection (Li et al., 2019).

### 3.2 Tracers advection term discrete schemes

Spurious diapycnal mixing is one of traditional errors in state-of-the-art atmospheric and oceanic model,

especially for the terrain-following coordinate regional models including both the continental slope and

deep ocean (Marchesiello et al., 2009; Naughten et al., 2017; Barnier et al., 1998). Marchesiello et al.

(2009) identified the problem of the erosion of salinity from the southwest Pacific model with steep reef

slopes and distinct intermediate water masses based on ROMS. They found that ROMS cannot preserve

the large-scale water masses while using the third-order upstream advection scheme during the spin-up

phase of the model, and proposed a rotated split upstream third-order scheme (RSUP3) to decrease

dispersion and diffusion by splitting diffusion from advection. They implemented RSUP3 by employing

a rotated biharmonic diffusion scheme with flow-dependent hyper diffusivity satisfying the Peclet

constraint.

For SCSOFSv1, a third-order upstream horizontal advection scheme (hereafter referred to as U3H), a

fourth-order centered vertical advection scheme (hereafter referred to as C4V), and the scheme of



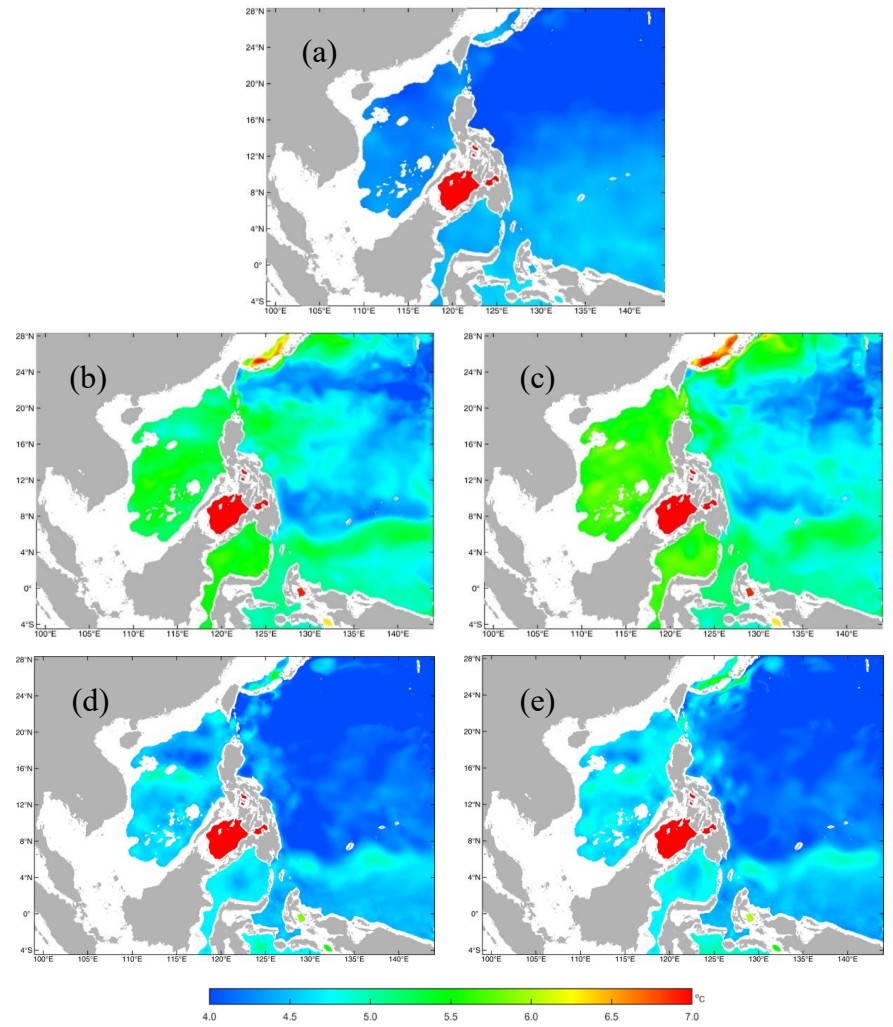

**Figure 5: The distributions of monthly mean temperature at 1000m layer in January from GDEMv3 climatology (a), the fifth (b) and the eleventh (c) model year by using the scheme combination of UCI based on SCSOFSv1 for other model settings, the fifth (d) and the eleventh (e) model year by using the scheme combination of AAG based on SCSOFSv2 for other model settings.**

horizontal mixing on epi-neutral (constant density, hereafter referred to as ISO) surfaces for tracers are

selected (Shchepetkin and McWilliams, 2005). We have encountered same problem with Marchesiello

et al. (2009) for temperature (Fig.5b and 5c) and salinity (Fig.6b and 6c) in deep layer. Figure 5 and 6

show the distributions of monthly mean temperature and salinity at 1000m layer in January from





GDEMv3 climatological initial fields, and the simulated results from the fifth and the eleventh model

years by using the scheme combination of U3H, C4V and ISO (hereafter referred to as UCI) and the

combination of the fourth-order Akima scheme (Shchepetkin and McWilliams, 2005) for both horizontal

and vertical advection terms and the scheme of horizontal mixing along Geopotential surfaces (constant

Z) for tracers (hereafter referred to as AAG), respectively, and other settings are identical with

SCSOFSv2. Figure 7 shows the comparisons of time series of domain averaged monthly mean

temperature and salinity at 1000 m layer simulated using the scheme combinations of UCI in SCSOFSv1

and AAG in SCSOFSv2, respectively. In order to save computation costs, we only run the model with

scheme combination of UCI for over 16 years till it reaches stable status.

The fourth-order Akima scheme is a little different from the fourth-order centered scheme by replacing

the simple mid-point average with harmonic averaging in the calculation of curvature term. Since the

time stepping is done independently from spatial discretization in ROMS, the Akima scheme represents

its advantage of reducing spurious oscillations, which arises with nonsmoothed advected fields, with

respect to the fourth-order centered (Shchepetkin and McWilliams, 2003, 2005).

During the spin-up phase of the model from the initial conditions derived from GDEMV3, the

temperature at 1000 m increases from 3.0-12.0℃ in initial (Fig.5a) to 3.0-17.2℃ (Fig.5b), and the

domain averaged monthly mean value quickly increases from 4.4 ℃ in initial to 5.1 ℃ (Fig.7a) in





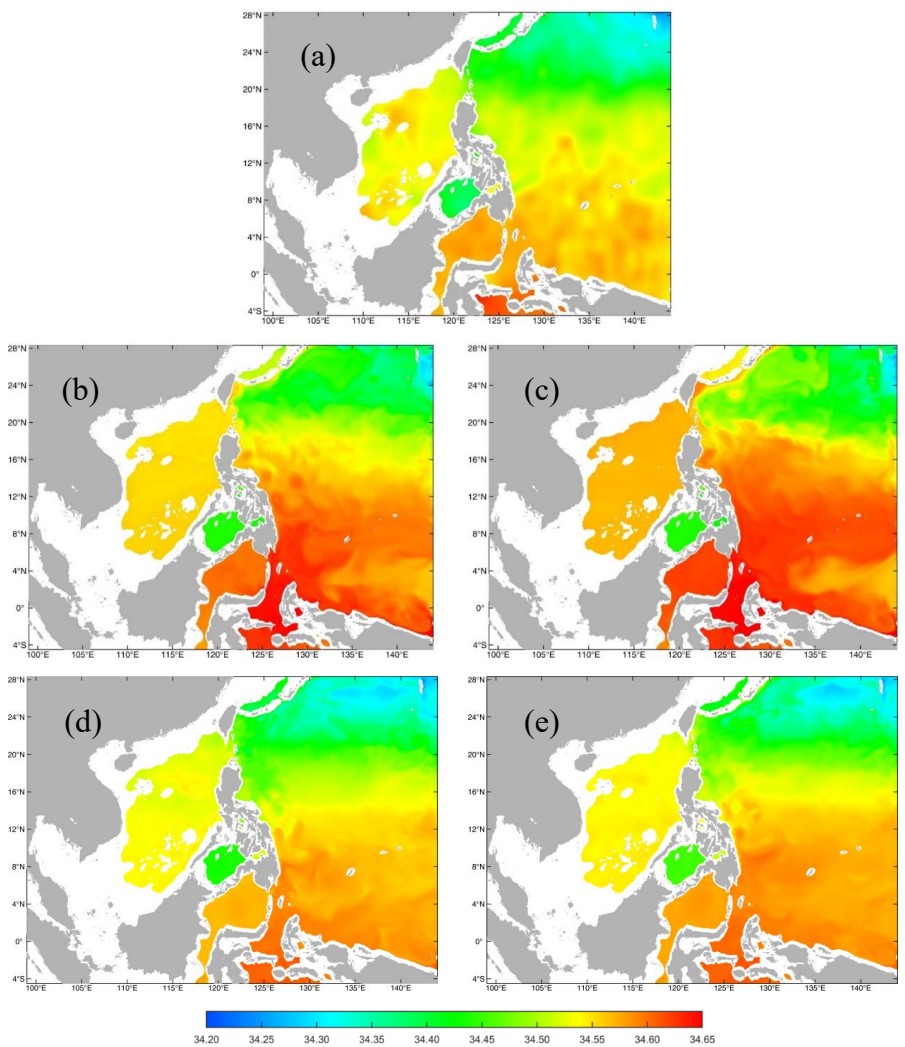

**Figure 6: The same as Fig.5, but for salinity.**

January of the fifth model year; the salinity at 1000 m increases from 34.26-34.62 in initial (Fig.6a) to

34.27-34.68 (Fig.6b), and the domain averaged monthly mean value increases rapidly from 34.50 in

initial to 34.54 (Fig.7b) in January of the fifth model year too. Especially, the increasing of domain

averaged monthly mean value is almost linearly for both temperature and salinity in the first 50 months,

indicating a fast increasing speed and strong spurious diapycnal mixing (Fig.7). Those values are even

higher in January of the eleventh model year, the ranges (minimum and maximum value) reach to 3.0-





17.3℃ for temperature (Fig.5c) and 34.26-34.73 for salinity (Fig.6c). The domain averaged values are

5.3 ℃ for temperature and 34.56 for salinity (Fig.7), respectively. The areas with increasing temperature

and salinity are mainly located at steep slopes and nearby regions, e.g. the central basin of SCS, the

Sulawesi Sea and the equatorial Pacific Ocean.

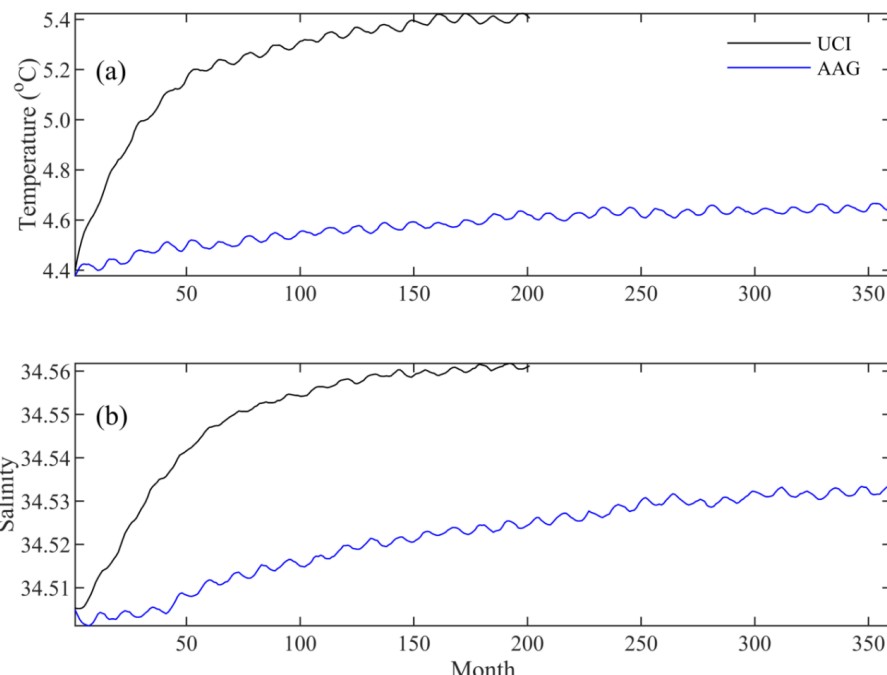

**Figure 7: The timeseries of domain averaged monthly mean temperature (a) and salinity (b) at 1000 m layer**

**simulated by using the scheme combinations of UCI (black line) and AAG (blue line), respectively**

To fix this problem, we tested various model settings and compiling options available in ROMS, such as

increasing the number of vertical levels, changing the advection and diffusion schemes, horizontal

mixing surfaces for tracers, horizontal mixing schemes. Details of how tested model settings effect on

the spurious diapycnal mixing are beyond the scope of this paper, which will be discussed in a separate

paper. Based on test results, we conclude that the spurious diapycnal mixing problem can be suppressed

significantly by employing AAG scheme combination (Fig.5d, e and Fig.6d, e) in SCSOFSv2.

The monthly mean temperature at 1000 m layer from SCSOFSv2 varies from 3.0-12.0℃ in initial

condition to 3.0-11.5℃ (Fig.5d), and the domain averaged monthly mean value increases slightly from

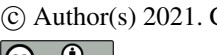



4.4 ℃ in initial to 4.5 ℃ (Fig.7a) in January of the fifth model year. The salinity at 1000 m varies from

34.26-34.62 in initial condition to 34.24-34.63 (Fig.6d), and the domain averaged monthly mean value

only slightly varies from 34.505 in initial to 34.509 (Fig.7b) in January of the fifth model year. Those

values show little variation till January of the eleventh model year, the ranges are 3.0-11.3℃ for

temperature (Fig.5e) and 34.25-34.63 for salinity (Fig.6e), and the domain averaged values are 4.6 ℃ for

temperature and 34.52 for salinity (Fig.7), respectively. For the increment of domain averaged values,

temperature is about 0.2℃ and salinity is about 0.03, yet remaining stable after 20 model years (Fig.7).

It is suggested that spurious diapycnal mixing has been suppressed significantly by AAG scheme

combination, which can preserve the characteristics of water masses in deep ocean well.

In addition, it is found that the model skill for SST has also been improved significantly while the new

AAG scheme employed in SCSOFSv2 (Fig.3 and Fig.4). The maximum of monthly mean differences

between simulated SST by SCSOFSv2 and OSTIA is about 3-4℃, which is obviously smaller than the

results from BulkFormula. The results of SCSOFSv2 show much more areas with lower SST than OSTIA

in the central Pacific Ocean, comparing to the results of SCSOFSv1 and BulkFormula, which can be

attributed to the new scheme combination. For the domain averaged RMSE of monthly mean SST of

SCSOFSv2 is about 0.65-0.84℃, with an annual mean value of 0.77℃, the maximum value (0.84℃) is

in January and December, the minimum value (0.65℃) is in May. Comparing with the results of

BulkFormula, the performance of model skill judging from the annual mean SST RMSE is improved by

about 23% due to employing new combination scheme in SCSOFSv2.

### 3.3 Data assimilation scheme

As mentioned as Zhu et al. (2016), the original SCSOFSv1 had employed the multivariate Ensemble

Optimal Interpolation (EnOI, Evensen, 2003;Oke et al., 2008) method to assimilate the along track

altimeter Sea Level Anomaly (SLA) data produced by SSALTO/DUACS and distributed by AVISO with

support from Center National D'études Spatiales. During this upgrading process, we also improved some

functions of EnOI scheme, and developed a new "Multi-source Ocean data Online Assimilation System"

(MOOAS).



Firstly, SCSOFSv1 assimilated the along track SLA data only, while SCSOFSv2 is additionally able to assimilate satellite AVHRR SST and in-situ temperature and salinity (T/S) vertical profiles data from the Argo arrays, simultaneously. It is conducted by constructing all innovations (difference between the assimilated observation and the model forecast), background error covariances, and observation errors for four different variables to one array, respectively. For the observation errors in SCSOFSv2, we simply

set those of SLA and SST as constants of 0.09 cm and 0.5 ℃, respectively; as for those of Argo T/S, assuming they are represented as a function of water depth ($D$) following Xie and Zhu (2010) as $ERR_T(D)=0.05+0.45exp(-0.002D)$, $ERR_S(D)=0.02+0.10exp(-0.008D)$.

Secondly, we have introduced the method of computing the anomalies of ensemble numbers used for constructing the background error covariance following Lellouche et al. (2013). In SCSOFSv1, the

anomalies are computed by subtracting a 10-year average from a long-term (typically 10 years) model free run snapshots with 5-day interval for the ocean state, i.e. sea surface height and three-dimensional temperature, salinity, zonal velocity, and meridional velocity. And the ensemble is selected within a 60 d window around the target assimilation date from each year, adding up to about 130 members in total (Ji et al., 2015; Zhu et al., 2016). However, in SCSOFSv2, a Hanning low-pass filter is employed to

create running mean according to Lellouche et al. (2013) in order to get intra-seasonal variability in the ocean state. Thus the anomalies are computed by subtracting the running mean with 20-day time window from a 10-year (2008-2017) free run daily averaged results. Especially, it is pointed out that the daily averaged free run results are selected within 60 d window around the target assimilation date from each year of 2008-2017 and used to compose ensemble members, thus about 590 members totally in

SCSOFSv2. It means that the background error covariances rely on a fixed basis and intra-seasonally variable ensemble of anomalies, which improves the dynamic dependency.



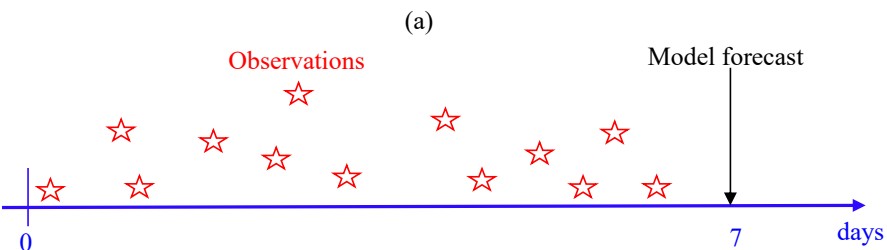

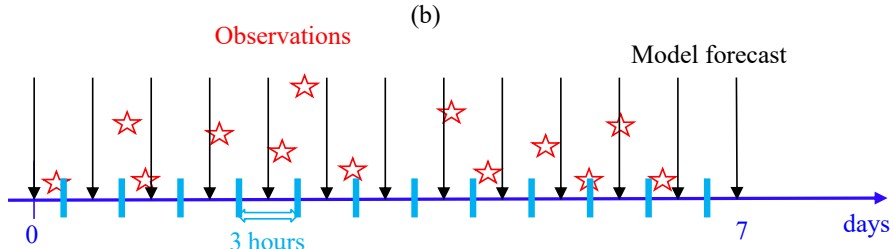

**Figure 8: Schematic representation of the FGAT method not used in SCSOFSv1 (a) and used in SCSOFSv2 (b). Red stars stand for observations, black arrows stand for archived snapshots of model forecast**

380 Thirdly, for each analysis step with a 7-day assimilation cycle, all observations of SLA within the 7-day time window before the analysis time are treated as observed at the analysis time in SCSOFSv1, with assumption of all observations were still valid at the analysis time. The time misfit between the observation and model forecast would cause non-negligible biases for the calculation of innovations. Actually, it is inconvenient to calculate the synchronous innovations between the observation and model

385 forecast entirely, since the spatio and temporal distributions of along-track SLA and Argo data are irregular and variable at each analysis step. In order to alleviate this deficiency, the First Guess at Appropriate Time (FGAT) method (Lee and Barker, 2005;Cummings, 2005;Lee et al., 22-25 June 2004;Sandery, 2018) is used in SCSOFSv2. Considering the intense computing and storage cost, we have divided the 7-day time window into 56 3-hour time slots (Fig.8), and archived 57 snapshots with a 3-

390 hour interval while model forecast run following the previous analysis run. Then the innovations can be calculated within each 3-hour time slot by using the observations subtracts the nearest model forecast. It means that the maximum temporal misfit of the innovations between the observation and model forecast





would be decreased from 7 days to 1.5 hours by using FGAT. Meanwhile, the localization is still used

with the radius set to be 150 km as in SCSOFSv1.

In SCSOFSv1, the analysis increments of sea surface height and three-dimensional temperature, salinity,

zonal and meridional velocities produced by each analysis of data assimilation are applied to the model

initial fields at one time step. It would induce model significant initial shock and spurious high-frequency

oscillation due to the imbalance between the increments and the model physics inevitably (Lellouche et

al., 2013; Ourmières et al., 2006), and usually causes a rapid growth of forecast error and even model

blow up after a few assimilation cycles or one or two-year period after the intermittent assimilation run.

It is a threat to the stability and robustness of OOFS. Therefore, we introduced the incremental analysis

update (IAU) method (Bloom et al., 1996; Ourmières et al., 2006) to apply each analysis increment to

the model integration as a forcing term in a gradual manner in SCSOFSv2 to diminish the negative impact.

In our case, we get the tendency term by dividing the increments with the total number of time steps

within an assimilation cycle as in most IAU methodologies, in order to make sure the time integral of

tendency term equals the analysis increment calculated by EnOI.

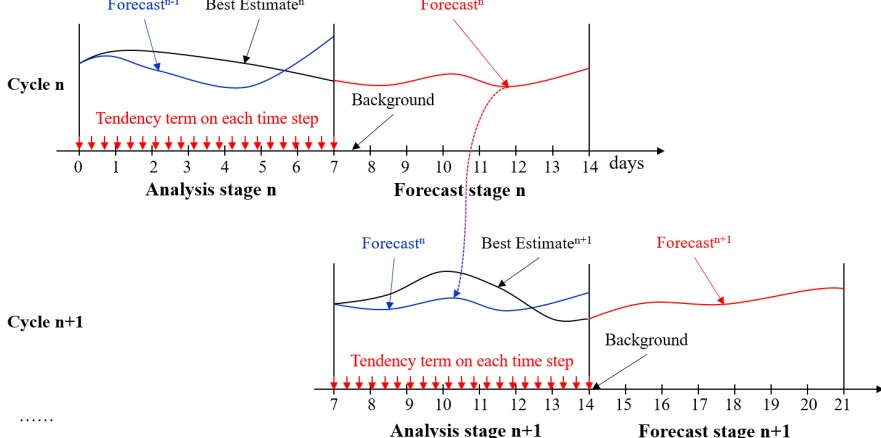

**Figure 9: Schematic representation of the data assimilation procedure for two consecutive cycles, n and n+1 in SCSOFSv2, while considering the FGAT and IAU methods.**



Once including the FGAT and IAU methods in EnOI scheme, the whole system integral strategy has to

be adjusted by adding one more model integration over the assimilation time window (Lellouche et al.,

2013). In SCSOFSv1, only one time model integration is needed. It means that once physical ocean

model finishes 7 days run (does not need to archive snapshot fields) and outputs a restart field, the EnOI

data assimilation module starts to calculate the analysis increments at the restart field time and adds it to

the restart field, then the physical ocean model makes a hot-start from the updated restart field to run 7

days for next cycle.

However, in SCSOFSv2, two times model integration is needed due to considering the FGAT and IAU

methods (Fig.8). It means that physical ocean model needs to be integrated 14 days in each assimilation

cycle, to add the tendency term to the model prognostic equations due to the IAU method used during

the first 7 days run (referred to as "Analysis Stage"), to output restart field at the end of 7th day for hot

starting ocean model in next cycle, and to output 3-hourly snapshots forecast fields during the second 7

days run (referred to as "Forecast Stage") to be used in next cycle by FGAT method. The model outputs

from the Analysis Stage are referred to as "Best Estimate", and from the Forecast Stage are referred to

as "Forecast". The analysis increments are defined at the 3.5th day, but not at the end of 7th day as in

SCSOFSv1, with the observed SLA and Argo T/S vertical profiles data within the 7-day time window

and AVHRR SST data on the 4th day used by FGAT method.

### 4.     Inter-comparison and accuracy assessment

In order to show the improvements of different SCSOFS sub-versions during the upgrading process, the

results of inter-comparison and assessment are shown in this section, by using the GOV Inter-comparison

and Validation Task Team (IV-TT) Class 4 verification framework (Hernandez et al., 2009). Class 4

metrics are used for inter-comparison and validation among different global or regional OOFSs or

assimilation systems originally (Ryan et al., 2015; Hernandez et al., 2015; Divakaran et al., 2015). It

includes four metrics, namely, bias for consistency, RMSE for quality or accuracy, Anomaly Correlation

(AC) for pattern of the variability and skill scores for the utility of a forecast. They are calculated



according to differences between model values and reference measurements in observations space for

each variable over a given period and spatial domain. The physical variables used in Class 4 metrics are

SST, SLA, T/S profiles, surface currents and sea ice. Reference measurements, providing ocean "truth",

are selected as follow, SST from *in-situ* drifting BUOY, SLA from AVISO along-track data, temperature

and salinity from Argo profiles, respectively. They are assembled by GOV IV-TT participating partners

on a daily basis (Ryan et al., 2015).

It is virtually impossible to test and validate exhaustively performances of all upgrades mentioned in

Section 2 and 3. Here, we separate the whole upgrading procedure from SCSOFSv1 to SCSOFSv2 into

four stages with three more sub-versions (v1.1, v1.2, v1.3) according to the reality. By respecting to the

previous version, the major upgrades in each new version are listed in Table1.

Table 1 The major upgrades with respect to the previous version

| SCSOFS versions | Settings updates |
|---|---|
| v1→v1.1 | **ROMS version** shifting from v3.5 to v3.7; **land-sea mask** redistribution; **bathymetry** substitution ETOPO1 with GEBCO_08; **initial T/S conditions** changing from SODA2.2.4 to GDEMV3; **open boundary data** changing from climatological monthly mean to monthly mean from 1990 to 2008 with SODA 2.2.4; **sea surface atmospheric forcing data** changing from NCEP Reanalysis 2 to CFSR; **the parameter dQ/dSST** changing from constant to temporal and spatial varying values; **sea surface atmospheric forcing method** changing from directly fluxes forcing to BulkFormula |
| v1.1→v1.2 | **Open boundary data** of SODA 2.2.4 monthly mean extending from 2008 to 2010; **the eastern lateral boundary** moving westward; **the observed SST data** for net surface heat flux correction changing from MGDSST to AVHRR |
| v1.2→v1.3 | Considering **mean seal level atmospheric pressure** effect, increasing **vertical layers** from 36 to 50; changing **the transform and stretching function**; **tracers advection discrete schemes** changing from UCI to AAG |
| v1.3→v2 | Including the MOOAS |



In this paper, we use Class 4 metrices and select the first four physical variables, SST, SLA and T/S, to inter-compare and assess the accuracy among different sub-versions of SCSOFS (Table 2). Since all the reference measurements data mentioned above have not been used in SCSOFS for those sub-versions

without data assimilation, they are independent reference observation from SCSOFS except for SCSOFSv2. The inter-comparison and validation among those sub-versions without data assimilation are conducted for the model free-run results in 2013, and between v1.3 and v2 are conducted in 2018 to validate the performance of MOOAS.

**Table 2 Mean values of each metric of the four physical variables for the best estimates of each sub-version**

| Variables | Metrics | v1 | v1.1 | v1.2 | v1.3 | | v2 |
|---|---|---|---|---|---|---|---|
| **SST** | AC | 0.52 | 0.56 | 0.58 | 0.62 | 0.64 | 0.74 |
| | Bias(℃) | 0.77 | 0.88 | 0.70 | 0.40 | 0.34 | 0.24 |
| | RMSE(℃) | 1.21 | 1.12 | 0.98 | 0.76 | 0.66 | 0.52 |
| **SLA** | AC | — | — | — | — | 0.67 | 0.85 |
| | Bias (cm) | -7.0 | -5.5 | -7.0 | -7.4 | -5.2 | -3.1 |
| | RMSE (cm) | 21.6 | 20.8 | 16.7 | 14.8 | 12.9 | 8.5 |
| **T Profile** | AC | 0.01 | 0.04 | -0.12 | 0.48 | 0.38 | 0.57 |
| | Bias (℃) | 0.98 | 0.75 | 0.30 | -0.15 | -0.08 | 0.15 |
| | RMSE (℃) | 1.75 | 1.60 | 1.44 | 1.03 | 0.96 | 0.67 |
| **S Profile** | AC | -0.01 | -0.02 | 0.02 | 0.44 | 0.30 | 0.51 |
| | Bias | 0.06 | 0.05 | 0.06 | 0.02 | 0.013 | 0.009 |
| | RMSE | 0.14 | 0.14 | 0.13 | 0.10 | 0.11 | 0.08 |
| **Year** | | | | 2013 | | 2018 | |

**4.1 SST**

The accuracy of SST is continuously increasing from version v1 to v2, with AC increased from 0.52 in v1 to 0.74 in v2 with percentage increase (PI) being 29.7%, RMSE is decreasing from 1.21℃ in v1 to 0.52℃ in v2 with PI being 57.0%, for the annual mean of whole model domain averaged in 2013 (or v1.3 and v2 in 2018) (Table 2). For the versions v1, v1.1, v1.2, v1.3, their AC show significant seasonal

variations, with high AC in summer and low AC in winter. It is also indicated that the accuracy of SST can be benefited from the sea surface atmospheric forcing method, more accurate observed SST data used for sea surface heat flux correction, temperature advection discrete scheme, and SST data assimilation.





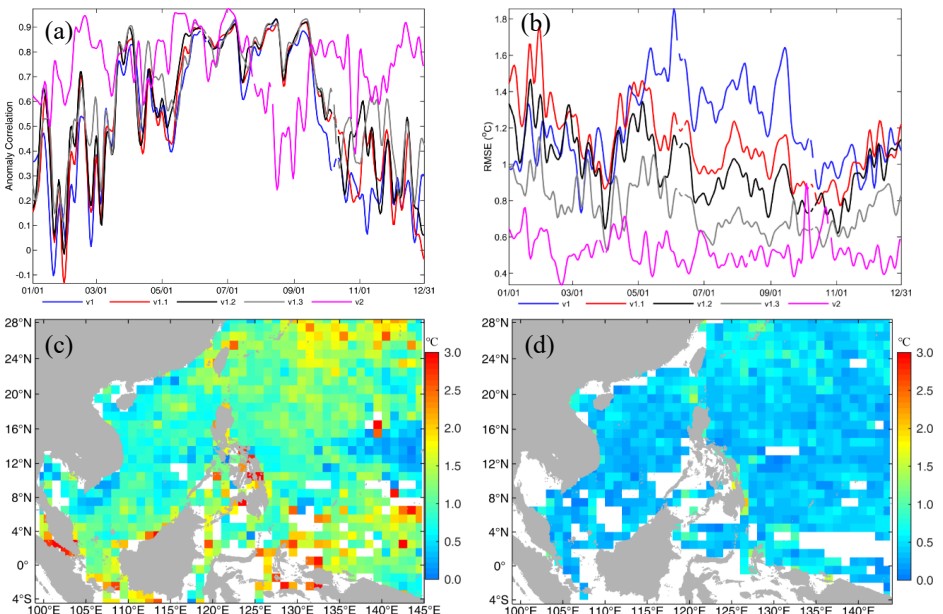

**Figure 10: Anomaly correlation (a) and RMSE (b) timeseries of SST best estimates for each version against observations as a function of time (7-day low pass filter applied), v1, v1.1, v1.2, v1.3 without data assimilation in 2013, and v2 with data assimilation in 2018. Horizontal distribution of SST RMSE in a 1°×1°bin for the version v1 (c) and v2(d), the calculation was performed for year round in 2013 and 2018, respectively**

The improvement of SST due to sea surface atmospheric forcing method changing mainly occurred in summer time, showing the same pattern as the result of the year 2014 in Fig. 3 and 4. But sea surface heat flux correction with more accurate observed SST data can improve accuracy of SST simulation for the whole year (v1.2 in Fig.10b). We also found that OISST data is closer to OSTIA than MGDSST (figure not shown). Due to benefit from those changes, the maximum and minimum value of SST RMSE have decreased from 1.92℃ and 0.71℃ in v1 to 1.52℃ and 0.60℃ in v1.2 for the whole year 2013, respectively. It is worth mentioning that AAG schemes combination not only improves the deep layer temperature, but also contributes to the improvement of SST due to internal baroclinic vertical heat transport. The maximum and minimum value of SST RMSE is 1.21℃ and 0.52℃ in v1.3. For the results with data assimilation in v2, the maximum and minimum value of SST RMSE is only 1.13℃ and 0.32℃, respectively. It is better than the result in v1.3 year-round.



For the horizontal distribution of SST RMSE, large values are mainly located at the areas near equator,

coast areas and northern lateral boundary, with most of values larger than 1.5℃ and maximum value

about 6.67℃ in v1 (Fig.10c). In v1.3, due to the contributions of all the above model updates, the pattern

of RMSE is similar with v1 basically without significant variations, but the maximum value decreases to

3.91℃ and most of values are less than 1.2℃. After applying MOOAS in v2 (Fig.10d), only a few large

RMSE values are located at the eastern coast of Philippine island with the maximum value of 2.09℃ and

most of values lower than 0.8℃. It indicates that the performance of SST in SCSOFSv2 has been

improved significantly due to all the updates mentioned above.

## 4.2 SLA

For the whole improving process, the accuracy of SLA is also continuously increasing from version v1

to v2, with RMSE decreasing from 21.6cm in v1 to 8.5cm in v2 with PI being 60.6%, for the annual

mean of whole model domain averaged in 2013 (or in 2018 for v1.3 and v2) (Table 2). Since there was

an ongoing problem with the SLA climatology variable provided by GODAE IV-TT during 2013-2015,

we could not calculate AC for SLA in 2013 and had feedbacked this issue to GODAE IV-TT. But from

the result of SLA AC in 2018, we can find that it increases from 0.67 in v1.3 to 0.85 in v2, showing

significant improvement for the correlation of pattern of the variability between the model results and

climatology.

From Fig.11(a), there is a slight decrease of RMSE in v1.1 with respect to v1, which mainly occurs in

winter time, and rarely in summer time. This maybe because no direct or intrinsic relationship between

those model updates from v1 to v1.1 and SLA in physics, and those updates mainly focus on horizontal

and temporal resolution of the datasets. However, the improvement of SLA accuracy is obvious in v1.2

with respect to v1.1, with the minimum and maximum of daily-mean RMSE values change from 0.12cm

and 0.31cm in v1.1 to 0.11cm and 0.23cm in v1.2, respectively. Their annual mean value decreases from

20.8cm in v1.1 to 16.7cm in v1.2, with PI of 19.7%. This may be resulted from well representing of NEC

pattern due to change of model eastern lateral boundary. With respect to v1.2, accuracy of SLA in v1.3

slightly increases with annual mean value 14.8cm and PI 11.4%. It may be resulted from the mean sea



level air pressure correction and modification of T/S baroclinic structures due to AAG employed. In addition, the most significant improvement for SLA is introduced by MOOAS, with minimum and maximum of daily-mean RMSE value are 6.1cm and 12.1cm in v2, respectively. The annual mean RMSE decreases to 8.5cm and PI reaches to 34.1% with respect to v1.3 and to 60.6% with respect to v1. It is

undoubtedly that this significant improvement is introduced by along-track SLA being assimilated into the system by MOOAS.

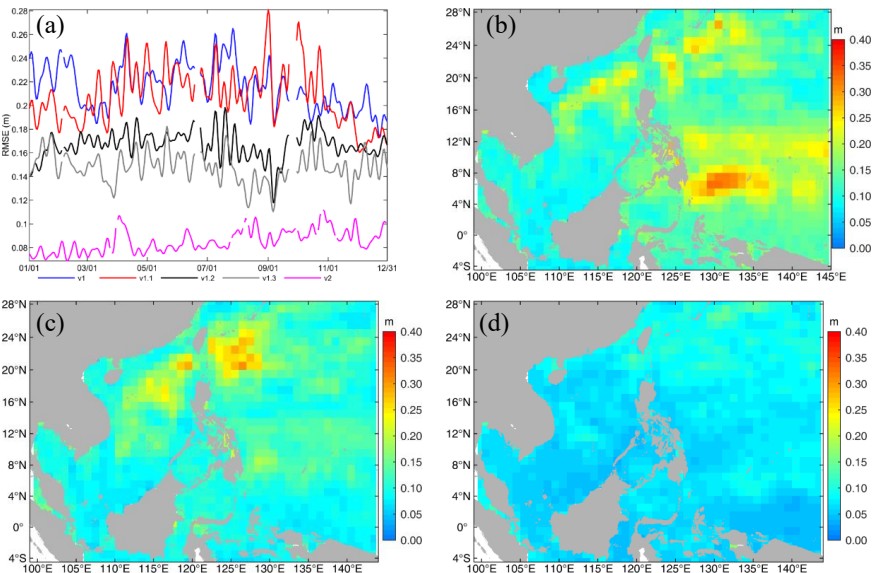

**Figure 11: (a) similar to Fig.10(b) but for SLA. (b), (c), (d) similar to Fig.10(c) or (d), but for SLA in v1, v1.3(in 2013) and v2, respectively.**

For the horizontal distribution of SLA RMSE, large values over 20 cm are mainly located in the area of NEC pathway, continental shelf of the northeastern SCS and northeast of LUS, with maximum value of 32.7cm in v1(Fig.11b). In v1.3 (Fig.11c), large values in the area of NEC pathway almost disappeared, the maximum RMSE is 30.3cm and most of values are less than 20cm, which may be interpreted as better representing of NEC pattern due to movement of model's eastern lateral boundary. By comparing with

v1.3 or even v1, SLA RMSE decreases dramatically for the whole model domain and does not show areas with obvious large values in v2, and its maximum value is only 18.2cm, with most of values less

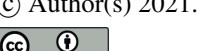

than 10cm. It is well known that plenty mesoscale eddies occur in each side of the Luzon Strait, northeastern SCS and western Pacific (Fig. 12a), large SLA RMSE in Fig. 11b and Fig. 11c indicating that pure physical ocean model cannot capture meso-scale process well without SLA data assimilated

(Fig.12b). However, Fig. 11d shows significant reduction with SLA RMSE, indicating that meso-scale eddies can be represented by SCSOFSv2 with along-track SLA data assimilated and agreement with satellite observations well (Fig. 12c).

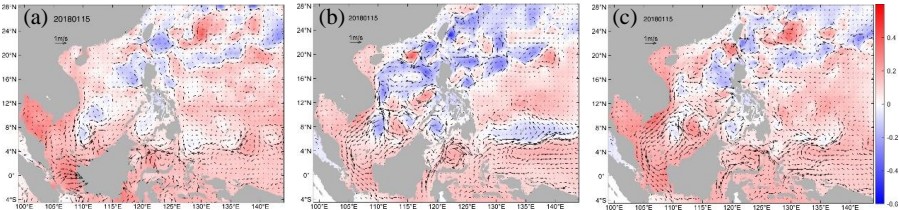

**Figure 12: Daily averaged SLA (color shaded) and surface velocity anomaly (vector) on January 15, 2018,**
**from AVISO, SCSOFSv1.3, and SCSOFSv2, respectively**

### 4.3 T/S profiles

For 3D T/S distribution, by comparing model results with climatology T/S profiles, the results from first three versions show poor correlation with observations (Fig.13a and Fig.14a) and large RMSE (Fig.13b and Fig.14b), i.e. 1.44-1.75℃ for T and 0.13-0.14 for S (Table 2), even if they decrease with model

updates. Especially, for the vertical distribution, the RMSE can reach to larger than 3℃ for T and 0.3 for S in thermocline and halocline, respectively, and remained larger than 1℃ for T in deep layer and 0.1 for S above 700m depth (Fig.13d and Fig.14d). This may result from spurious diapycnal mixing due to UCI schemes combination employed. Those updates in v1.1 and v1.2 can only slightly improve 3D T/S, and cannot contribute to their intrinsic improvements, neither for surface forcing nor for lateral boundary

conditions, except for surface layer of shallower than 100m.

However, once AAG schemes combination employed in v1.3, the improvements to 3D T/S are obvious with respect to the first three versions (Fig.13a,b and Fig.14a,b). The AC increases to 0.38-0.48 for T and 0.30-0.44 for S, and RMSE decreases to 0.96-1.03℃ for T and 0.10-0.11 for S, respectively (Table 2). For the vertical distribution, the AC remains around 0.4 for both T and S in the whole water column, and over 0.6 for T in the surface layer (Fig.13c and Fig.14c), RMSEs significantly decrease to less than






2℃ for T in thermocline and 0.25 for S in halocline, and less than 1℃ for T and 0.1 for S in deep layer

(Fig.13d and Fig.14d).

For the horizontal distribution of 3D T/S RMSE, RMSE of T is more likely being more than 1.5℃ with

maximum and minimum values being 4.45℃ and 0.49℃ (Fig.13e), and RMSE of S is larger than 0.1,

with

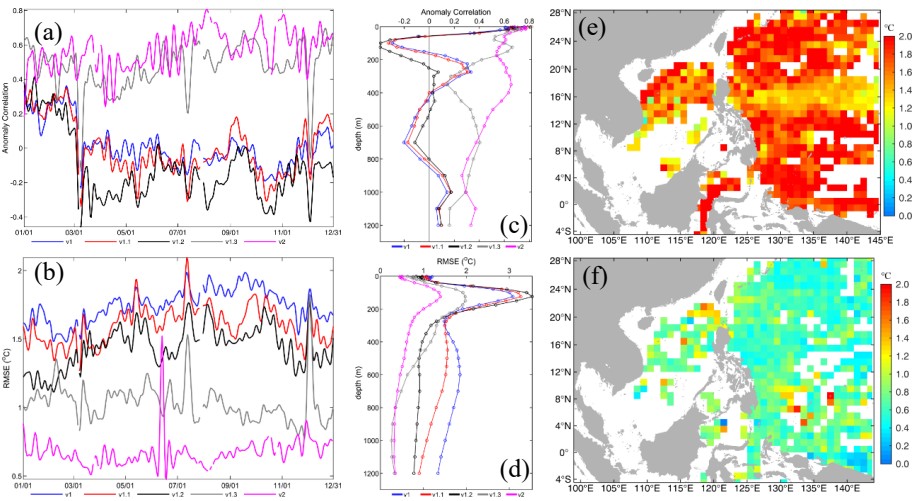

**Figure 13: (a) and (b) similar to Fig.10(a) and (b) but for T profile, respectively. (c) and (d) ) vertical distribution of best estimates for each sub-version against observations as a function of depth, v1, v1.1, v1.2, v1.3 without data assimilation in 2013, and v2 with data assimilation in 2018. (e) and (f) similar to Fig.10(c)**
**and (d), but for T profile in v1 and v2, respectively.**

maximum and minimum values being 0.81 and 0.06 (Fig.14e), in v1. Large values for S mainly locate in

the SCS and near equator in the Pacific Ocean. The trend is same with timeseries of RMSE, the horizontal

distribution of T/S RMSE shows slight decrease from version v1 to v1.2, but dramatic decrease in v1.3

(Figures not shown). Since it is benefited from AAG schemes combination in v1.3, most of T RMSE is

lower than 1.0℃, with maximum and minimum values being 1.72℃ and 0.11℃, and most of S RMSE

is less than 0.1 with maximum and minimum values being 0.62 and 0.03 in 2013, respectively.

By employing MOOAS, accuracy of 3D T/S has been improved continuously in v2 compare to v1.3 for

all the metrics in 2018 (Fig.13 and Fig.14). The mean AC has increased from 0.38 to 0.57 for T, and



from 0.30 to 0.51 for S. The mean RMSE has decreased from 0.96°C to 0.67°C for T, and from 0.11 to

0.08 for S (Table 2). For vertical distribution of AC for T, it's over 0.6 in surface, over 0.4 above 600m,

and over 0.3 in deep layer (Fig.13c). RMSE of T is less than 1.5°C for the whole vertical profile, and the

maximum value is located at the thermocline similar with other versions, but the error decreases

dramatically (Fig.13d). Unlike T, vertical AC of S does not show significant improvement in v2 with

respect to v1.3 below 200m, and it shows a little higher than which in v1.3 (Fig.14c) in above 200m. S

RMSE is less than 0.25 for the whole vertical profile, with the maximum value located at surface and

decreasing with depth, and decrease to less than 0.05 below 600m (Fig.14d).

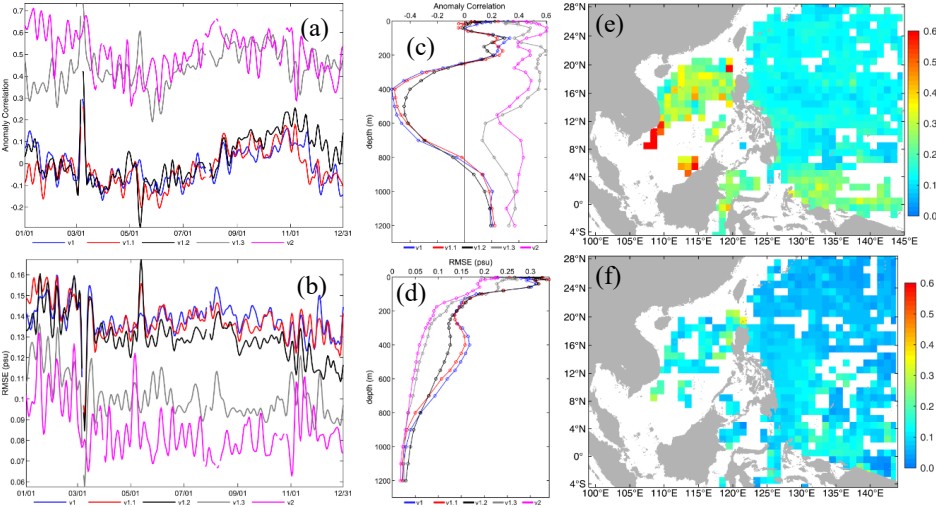

**Figure 14: Similar to Fig.13, but for S profile.**

For the horizontal RMSE distribution in v2, most of T RMSE is larger than 0.8°C with maximum and

minimum values being 1.96°C and 0.03°C (Fig.13f); and most of S RMSE is greater than 0.1, with

maximum and minimum values being 0.35 and 0.01 (Fig.14f), respectively, in 2018.

## 5.    Conclusions

This study illustrates major updates applied on SCSOFSv1 in both physical model settings, inputs and

EnOI data assimilation scheme in recent couple years following the recommendations of Zhu et al. (2016),



such as land-water grid mask redistribution, data sources for bathymetry, initial condition, sea surface

forcing and open boundary condition changing to higher spatial and temporal resolution, moving the

eastern lateral boundary westward, increasing vertical layers of model, and so on.

Three most significant updates are highlighted in this paper. Firstly, sea surface atmospheric forcing

method has been changed from direct forcing to BulkFormula to acquire effective negative feedback

mechanism of air-sea interaction by using COARE3.0 bulk algorithm. Upgrades lead to more reasonable

SST simulation with eliminating of abnormal values, significantly dropping of the maximum value of

monthly mean differences between simulated SST and OSTIA, and decreasing of domain averaged

RMSE of monthly mean SST from 0.99-1.62°C in SCSOFSv1 to 0.87-1.15°C in BulkFormula run. The

annual mean value decreases from 1.27°C to 1.00°C, indicating that the performance of model skill has

improved by about 21%.

Secondly, tracers advection term discrete scheme UCI has been substituted with AAG in order to

suppress spurious diapycnal mixing problem. After this substitution, the domain averaged monthly mean

temperature at 1000m layer decreases from 5.1°C to 4.5°C, and which of salinity decreases from 34.54

to 34.509, in January of the fifth model year, respectively. Even after 20 model years, domain averaged

values of temperature and salinity increments are about 0.2°C and 0.03, suggesting that AAG schemes

combination can well preserve the characteristics of water masses in deep ocean. In addition, model skill

for SST also can benefit from AAG schemes combination with annual mean domain averaged RMSE

decreasing from 1.00°C to 0.77°C, showing 23% improving rate for the performance.

Thirdly, the original EnOI method in SCSOFSv1 has been upgraded to new MOOAS by adding four new

functions. The multi-source observation data (SST, SLA, and Argo T/S profiles) can be assimilated

simultaneously; Hanning high-pass filter is applied to ensemble members from 10 years free run while

calculating the background error covariances to improve the dynamic dependency; FGAT method with

3-hour time slot is used to calculate the innovations; and IAU technique is employed with 7-day time

window to apply analysis increment to the model integration in a gradual manner.

Moreover, inter-comparison and accuracy assessment among five versions are conducted based on GOV

IV-TT Class 4 metrics for four physical variables, SST, SLA, and T/S profiles. The improvement of



accuracy of simulated SST mainly attributes to more accurate observed SST data source used for sea surface heat flux correction, BulkFormula method for sea surface atmospheric forcing, AAG temperature advection discrete scheme. The improvement of SLA accuracy mainly benefits from good

representations of NEC pattern caused by modification of model eastern lateral boundary, mean sea level air pressure correction, and T/S baroclinic structures improvement due to AAG employed. The improvement of 3D T/S mainly benefits from AAG non-spurious diapycnal mixing schemes combination employed.

At last, remarkable improvements for all above four variables are also benefited from MOOAS

application. With respect to v1.3, domain averaged annual mean SST RMSE decreases from 0.66℃ to 0.52℃ with PI being 21.2%, SLA RMSE decreases from 12.9cm to 8.5cm with PI being 34.1%, T profile RMSE decreases from 0.96℃ to 0.67℃ with PI being 30.2%, S profile RMSE decreases from 0.11 to 0.08 with PI being 27.3%, in v2 while using MOOAS.

Although SCSOFSv2 has improved greatly comparing to the previous versions, some biases still exist in

surface and subsurface. We plan to continue to improve the system in both physical model settings and data assimilation scheme for next step, such as sub-grid parameterization scheme for unresolved physical processes, vertical turbulent mixing scheme to consider wave mixing, more accurate input and forcing data source, and assimilating more or new types of observations (glider or mooring T/S, drifting buoys, *in-situ* velocity from moorings) into the system.

*Code and Data availability. The latest version of the source code for EnOI and ROMS trunk used to producethe results in this paper can be accessed via [https://doi.org/10.5281/zenodo.5215783](https://doi.org/10.5281/zenodo.5215783).* GEBCO_2014 Grid, [https://www.bodc.ac.uk/data/open_download/gebco/GEBCO_30_SEC/zip/](https://www.bodc.ac.uk/data/open_download/gebco/GEBCO_30_SEC/zip/), last access 3 January 2021; SODA 3.3.1, [https://www2.atmos.umd.edu/~ocean/index_files/soda3.3.1_mn_download.htm](https://www2.atmos.umd.edu/~ocean/index_files/soda3.3.1_mn_download.htm), last access 3 January 2021; SODA3.3.2, [https://dsrs.atmos.umd.edu/DATA/soda3.3](https://dsrs.atmos.umd.edu/DATA/soda3.3).2/REGRIDED/ocean/, last access 3 January 2021; CFSR, [http://rda.ucar.edu/datasets/ds093.0](http://rda.ucar.edu/datasets/ds093.0),

last access 3 January 2021; CFSv2, [http://rda.ucar.edu/datasets/ds094.0](http://rda.ucar.edu/datasets/ds094.0), last access 3 January 2021; NCEP_Reanalysis 2, [https://www.psl.noaa.gov/data/gridded/data.ncep.reanalysis2.html](https://www.psl.noaa.gov/data/gridded/data.ncep.reanalysis2.html), last access 3 January 2021; AVHRR, [http://www.ncei.noaa.gov/data/sea-surface-temperature-optimum-interpolation/v2.1/access/avhrr/](http://www.ncei.noaa.gov/data/sea-surface-temperature-optimum-interpolation/v2.1/access/avhrr/), last access 3 January 2021; OSTIA, SST of *in-situ* drifting BUOY, AVISO along-



track SLA, and Argo temperature and salinity profiles, https://marine.copernicus.eu/, last access 3
January 2021.

*Author Contributions.* XZ performed the physical model improvement and free-run simulations,
designed and wrote the paper. XZ and ZZ updated MOOAS and performed the data assimilation
simulations. SR and AL analysed and assessed model results. SR, HW and YZ helped in reading and
commenting on the paper. MZ helped in polishing the paper.

*Competing interests.* The authors declare that they have no conflict of interest.

*Acknowledgements.* This work was supported by the project of Southern Marine Science and
Engineering Guangdong Laboratory (Zhuhai) (No. SML2020SP008), "The Program on Marine
Environmental Safety Guarantee" of "The National Key Research and Development Program of
China" (No. 2016YFC1401605) and the National Natural Science Foundation of China (NO.
41806003). We would like to thank the anonymous reviewers for their careful reading of the
manuscript and for providing constructive comments to improve the manuscript.

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
