# Peer review of "Improvements in the regional South China Sea Operational Oceanography Forecasting System (SCSOFSv2)"

_Geoscientific Model Development, 2021_

## Author Comment (AC1)

**Comment on gmd-2021-235**
Anonymous Referee #1

Summary:
The manuscript represents updates of a South China Sea Operational Oceanography Forecasting System from version 1 to version2, which provide daily updated hydrodynamic forecasting in the South China Sea for the future 5 days. Comprehensive updates of model configuration and assimilation schemes have been mentioned. Among them, three major changes have been highlighted, namely the way of prescribing buoyancy flux, the tracer advection discrete scheme and data assimilation scheme. The model shows enhanced performance in the accuracy of the sea surface temperature and sea surface height.
Ocean states prediction has always been a challenging task and is of vital importance to the hazard prevention such as tropical cyclones and internal waves and so on. The South China Sea has unique monsoon system and topography and external Kuroshio forcing, which masks it a challenging region for ocean prediction. This paper is generally well written, and the way of improvement is well presented, which makes the comparison of results quite convincing. I believe this manuscript can be the basis of a useful publication after minor improvements.

The authors thank the reviewer for the insightful comments, and we completely agree with the questions and comments raised by the reviewer, which have helped us to improve the quality of the manuscript. We have carefully considered the reviewer's comments. Detailed replies to specific comments by the reviewer are presented below:

1 There are too many acronyms and I sometimes have to go back to look for its meaning. I would recommend keeping some acronyms frequently used by other articles, such as SCS and OFS and SST, but don't use abbreviation for only two words (such as RTOFS, PI) and abbreviation that is too hard to recall (such as RSUP3, U3H and C4V).

Line46: remove (GODAE);

Line52: remove (CONCEPTS);

Line54: remove (RTOFS)

Line56: remove (HYCOM)

Line58: remove (JMA)

Line61: remove (WNP)

Line62: remove (CGOFS)

Line75: remove (3DVAR)

Line77: remove (4DVAR)

Line86: remove CMEMS, and ; Lellouche et al.

Line172: remove (JMA)

Line287: remove (RSUP3)

Line289: replace RSUP3 with the rotated split upstream third-order scheme

Line291: remove (hereafter referred to as U3H)

Line292: remove (hereafter referred to as C4V)

Line298: remove ,hereafter referred to as ISO

Line303: replace U3H with third-order upstream horizontal advection, replace C4V with fourth-order centered vertical advection, replace ISO with horizontal mixing on epi-neutral surfaces

Line518, 519: replace GODAE with GOV

2 Line 1: The improvements to the regional South China Sea Operational Oceanography Forecasting System include all the previous versions. I suggest removing the acronym of "(SCSOFSv1)".

Thanks for your pointing out this. But the acronym of "(SCSOFSv1)" and its version number are added following with the editor's review before public discussion.

3 Line 199: Please clarify the wards in this headline. What are the highlights and sensitive updates respectively?

We think the three improvements mentioned in this section are most important than others mentioned in section 2. They have significantly improved the model skill of SCSOFS from different aspects. And we have added some explanation it in Line 212.

4 Section 3.1, Line 220: The use of bulk formulation to calculate the buoyancy fluxes is reasonable, but it is not a real negative feedback because the atmospheric forcing, such as air temperature, relative humidity are prescribed, which are not adjusted from the modelled SST. Please clarify this sentence, e.g. how the SST is improved through the use of bulk formulation should be further elaborated.

Thanks for pointing out this. We agree that the use of bulk formulation does not represent a real negative feedback like ocean-atmosphere coupled model, since the atmospheric forcing is prescribed. But it still can play a role with negative feedback function to the simulation of SST, because the calculation of sensible heat flux, latent heat flux and longwave radiation uses SST calculated by ocean model. We have added a reference Li et al. (2021) to elaborate the calculation of three air-sea fluxes in Line 230 and Line 757.

5 Figure 3. For the SCSOFSv1, the area north of 24°N in the BulkFormula is even warmer than that in the no BulkFormula experiment. But the SST is much improved in the SCSOFSv2. Later results in Section 3.2 indicate that it may be related to the improved advection and mixing scheme. Please further explained this by providing more information.

Thanks. We think this should be considered as two different problems separately. For the first one, the area north of 24°N in the BulkFormula is even warmer than that in the no BulkFormula experiment, it should be attributed to the local complicated air-sea interactions in the area and tidal mixing is missing in the model. For the second one, the SST is much improved in the SCSOFSv2, it mainly due to the improved advection and mixing scheme representing the vertical heat transport well, then the surface layer affected by subsurface and deep layer processes. We have added more explanations in Line 274, Line 278, and Line357-Line360. We also changed the title of figure 3 from SCSOFSv2 to SCSOFSv1.3, it was from the model result without data assimilation.

6 Section 3.2: What about the temperature and salinity bias in the subsurface layer in

the AAG scheme combination?

Please refer to Figure 13 and 14 for the results from v1.3. We have modified it in Line 365.

7 Line 335. Please explain the improvement of temperature and salinity with more detail. What is the diffusion term and advection term look like in the AGG scheme combination?

Thanks. The harmonic mixing scheme is used for both viscosity for momentum and diffusion for tracers in horizontal. And Mellor-Yamada Level-2.5 vertical mixing closure scheme is used for both momentum and tracers. This has been explained in Line 194-Line199.

8 Line 360: Why do you set observational error for the SLA and SST as 0.09 cm and 0.5 °C? The along-track satellite data contains high-frequency noise, especially for the shallow area (Zhou et al., 2015). How do you filter out this noise?

Zhou, Xiao-Hui, Dong-Ping Wang, and Dake Chen. "Global wavenumber spectrum with corrections for altimeter high-frequency noise." Journal of Physical Oceanography 45.2 (2015): 495-503.

Thanks for pointing out this issue. We have revised this in text Line 378-Line386.

We have used filtered SLA for assimilation products specially from AVISO, which is filtered with 20-day cutoff-period but not subsampled unlike other L3 products.

9 Line 428: In the section 4, I suggest introducing why the SST, SLA and T/S profiles are used to validate the model. Are those element enough to represent the outputs from the model? In addition, you provide the importance of SST in Line 210 to connect with the sea surface atmospheric forcings, but what kind of validations is related to section 3.2 and 3.3.

Thanks. We employ the IV-TT Class4 verification framework to validate the model, which is an international common verification metric. The main reason for using the SST, SLA, and T/S profiles is that public data are easy to obtain and has accumulated plenty of data. Of course, these elements are not enough to represent the outputs from model. But we can not get enough currents observation data to validate our model. Validations with the subsurface layer temperature and salinity using T/S profiles mainly relate to section3.2, all those validations are related to section 3.3, since all those three kinds of data have been assimilated by MOOAS.

10 Table 1: In Line 145 you mentioned the the new SODA 3.3.1 and 3.3.2 reanalysis were used, but in Table 1 you still mention the SODA 2.2.4, please check all the settings in this table.

Thanks. We have added one more information in new Table 2, as "Changing the **open boundary data** from SODA 2.2.4 monthly mean to SODA 3.3.1 and 3.3.2" while upgrading from v1.2 to v1.3.

11 Figure 13: Why the RMSE of temperature is suddenly large in June in Fig. 13b? In the paper of a recently published paper (Ding et al., 2021), there is also a similar large bias in June, can you provide some explanations to this?

Ding, R., Xuan, J., Zhang, T., Zhou, L., Zhou, F., Meng, Q., and Kang, I. (2021), Eddy-Induced Heat Transport in the South China Sea, Journal of Physical Oceanography, 51(7), 2329-2349, doi: 10.1175/JPO-D-20-0206.1.

Thanks for pointing out this issue. We have checked the original data in detail, and found that there were two Argo temperature profiles with bad quality as shown in following figures. The temperature observation is almost 40 °C at surface, and more than 10°C below 1000m layer. We have removed the two profiles, recalculated RMSE and replotted the figure 13b. We are sorry about this.

[Figure]

12 Please check the font size in all the figures to make sure it is clear. Also add title for similar plots.

Thanks. We have replotted the figure 2, 3, and 12 with larger font size.

---

## Author Comment (AC2)

**Comment on gmd-2021-235**

**Anonymous Referee #2**

The authors describe the evolution of the South China Sea operational oceanography forecasting system (SCSOFS), which significantly improve the quality of the system. The manuscript provides a detailed description of different versions of the ROMS model which were used for SCSOFSv1 and SCSOFSv2 and compare the 11-year long free run of both models. The results show various improvements for the changes that have been made to the operational system. It provides useful suggestions for the community. My opinion is that the paper is worthy of publication in GMD after minor modifications. But several specific aspects of the paper detail and analysis should be considered first.

Thanks for your confirmation to our manuscript.

**2** Specific comments**

The grammar and quality of the writing needs to be improved before publication: many sentences are over-complicated or confusing (Example lines: 113-114, 170-174, 341-343....). It would benefit from a thorough revision by someone with proficient written English skills.

We have invited a professional of similar academic background with proficient English written skills to proofread and revise our manuscript thoroughly according to the reviewer's comments.

Line116: "sea areas" change to "areas"

Line341-343: "The results of SCSOFSv2 show much more areas with lower SST than OSTIA in the central Pacific Ocean, comparing to the results of SCSOFSv1 and BulkFormula, which can be attributed to the new scheme combination." change to "Comparing with the results of SCSOFSv1 and BulkFormula, smaller SST hot bias versus OSTIA is found in the central Pacific Ocean for the results of SCSOFSv1.3, which can be attributed to the new scheme combination."

Line 99: the section title is not clear, which datasets? Not all datasets used in the article are described here.

Thanks. We have changed to "input datasets" in Line102.

Line 142: For what reasons are the initial conditions not deduced from soda 3.3 in correspondence with the boundary conditions?

Thanks. One reason is that the resolution of GDEMV3  $(0.25^{\circ} \times 0.25^{\circ})$  is finer than SODA3.3  $(0.5^{\circ} \times 0.5^{\circ})$ , and GDEMV3 is derived from plenty of observations with good accuracy. The other reason is that our model was started from climatological run with climatological initial, boundary, and surface forcing conditions. We just selected climatological initial conditions with finer resolution.

Lines 180-184: Why does the inclusion of Guam at the open boundary have such a

significant influence on the latitude of the NEC in the model? This need be explored or explained.

Thanks. It is because that the Guam Island changes the bathymetry from over 3500m to below 500m suddenly, and as a big block to the NEC once flowing into the model domain from eastern lateral boundary. We have explained it in Line 179.

Lines 189-193: Why these choices as opposed to the ones used in v1? What is special about 3rd order and biharmonic?

Thanks for pointing out this. I have clarified it between Line194 and Line199, and added one more table1. Actually, the third-order upstream and fourth-order centered schemes are used for momentum in SCSOFSv1, which are default and recommended settings in ROMS. We do not use biharmonic mixing scheme, just harmonic mixing for both SCSOFSv1 and SCSOFSv2.

Line 227: Any reference for OSTIA first appeared in the text? Is it provided to SCSOFS directly by the Metoffice or by other means?

In our manuscript, we have already cited the paper of Donlon et al.,2012 in the end of this sentence. OSTIA data is open access to public. We have already change 'provide' to 'produced' in Line 242 to avoid possible misunderstanding.

Line 326: the sentence on the conclusion may come after the lines 327-337.

Thanks, we have deleted the conclusion in Line339

Line 355: Where do you get the data from in SCSOFSv2, is it still the same servers? Where do the ARGO observations come from?

Thanks. Yes, we have got all SLA, OSTIA, ARGO observation data from the website https://marine.copernicus.eu of CMEMS in SCSOFSv2, it is the same with SCSOFSv1. Line 373: "around" is not an exact annotation, need to point out exactly.

Thanks. We have changed it to ", with 30 days before and after" in Line 399.

Line 383: "for the calculation of innovations" change to "when calculating innovations" Thanks. We have done it in Line 409.

Line 385: Lee et al., 22-25 Jun, format error!

Thanks. We have removed "22-25 June" in Line 414.

Line 619: What are the known biases of the system? Some are described in the article; can you summarize them in the conclusion? Can you elaborate the summary in terms of what is the added value of this regional system compared to existing global systems? Thanks for your good suggestions. We have added it in conclusions section. For the comparison to existing global system, it is not done in this article, so we can not give more information. We will try to do that with many existing global systems in later.

Section 2: A table summarizing the v2 and v1 features would be useful.

Thanks for your good suggestions. We have added a new table1.

Fig. 8: Why 7-day FGAT? Why not 3-days or 10-days? Can you explain more on this? The length of First Guess at Appropriate Time (FGAT) is associated with the data assimilation window, which is taken as 7 days in our data assimilation system. The length of assimilation window is decided by several factors. 1) The observations of Sea Level Anomaly (SLA) and temperature/salinity profiles are relatively sparse, therefore,

the window is enlarged enough to include more observations. 2) the true state evolves with time, therefore, the window is shortened, as the difference between observation and background increases with time. 3) The data assimilation system runs every 7 days, and forecasting system provides the forecast of 7 days in future, therefore, 7 days of window is adopted to collect the more observations, and also to avoid reusing some observations. The window of 7 days is also used by other operational oceanography system, such as the PSY4 in Mercator. Therefore, the length of 7 days is appropriate for the assimilation window, and also for the FGAT.

---

## Author Response (AR2)

**Comment on gmd-2021-235**

**Responding to Topical Editor**

The manuscript has been clearly improved. However, there are still some major issues to be solved before it can be accepted for publication. First, very importantly, English writing in the manuscript needs to be thoroughly checked, including grammar issues and uncommon representations.

We have invited a professional of similar academic background with proficient English written skills to proofread and revise our manuscript thoroughly according to the reviewer's comments.

Line16: future changed to next;

Line18: It changed to The paper;

Line70: need to be updated and improved continuously *changed to* has been upgrading and improving constantly

Line72: from a simple one to sophisticate one *changed to* into a more sophisticated level

Line73: amounts and types changed to diverse source

Line77: is changed to was

Line84: update changed to updates

Line104: contribution of one changed to contributions of each

Line116: remove sea

Line165: So changed to Therefore

Line175: NEC separated changed to NEC is separated

Line185: get changed to grow

Line186: removed changed to got removed

Line187: and would better be set to far away from island enough *changed to*, and it would be better if being set to far enough away from island

Line95: ,respectively removed

Line203: and in section 3 changed to and to be introduced in Section 3

Line211:, that are not mentioned in Sect.2, changed to we elaborate solutions to such

problems applied in SCSOFSv2, which has not been mentioned in Section 2

Line251: for changed to in

Line254: hard to be resolved changed to nearly impossible to resolve

Line256: more changed to higher amount of

Line257: heating up changed to heating

Line275: does not work better than direct forcing at all time *changed to* is not necessarily a panacea

Line318: in initial changed to by initial settings

Line319: in initial removed

Line323: in initial changed to by initial settings

Line325: in initial removed

Line327: increasing changed to increase

Line402: thus about 590 members totally changed to thus totally about 590 members

Line425: induce model significant initial shock and spurious high-frequency oscillation due to *changed to* induce a significant initial shock and spurious high-frequency oscillation to the model due to

Line428: even model blow up changed to even lead to model blowing-up

Line440: only one time model integration is needed *changed to* model integration is needed only one time

Line441: 7 days changed to a 7-day

Line446: two times model integration is needed *changed to* model integration is needed twice

Line453: referred to changed to reffered

Line489: show changed to shows

Line491: more accurate observed SST data used for *changed to* usage of more accurate observed SST data for

Line519: improving changed to upgrading

Line537: AAG employed changed to AAG being employed

Line550: be interpreted as better representing of NEC pattern due to movement of *changed to* be interpreted as a better representing of NEC pattern due to amendment of Line572: except for *changed to* with an exception of

Line617: on SCSOFSv1 in both changed to to SCSOFSv1 in aspects of

Line621: moving changed to shifting

Line625: significantly changed to significant

Second, the number of acronyms should be further reduced. It is really hard to remember them when reading the paper. Please reduce the use of acronyms to a lowest possible level. Just two examples, AC and PI, it would be much easier for readers if you write out the full words instead of using their acronyms. Wherever possible, I would suggest to use full words.

Thanks for pointing out it for us. We also have recognized this problem, and revised the manuscript as follow,

Line37, 39: LUS changed to Luzon Strait

Line55: remove NOAA

Line78: remove (MOI)

Line83: MOI changed to Mercator Ocean International

Line129: remove (NGDC)

Line145: remove (T/S)

Line146: Version changed to version, GDEMV3 changed to GDEMv3

Line157: Version changed to version

Line158: remove (NCAR)

Line168: remove (NCEI)

Line172: NP changed to North Pacific

Line245: OSTIA SST changed to OSTIA

Line250, 251: SST removed

Line377: (T/S) removed

Line462: Anomaly Correlation (AC) changed to anomaly correlation

Line465: T/S changed to Argo

Line475: T/S changed to Argo profiles

Line483: (T denotes temperature, S denotes Salinity, AC denotes anomaly correlation) *added*

Line485: AC changed to anomaly correlation

Line486: (PI) removed

Line487: PI changed to percentage increase

Line489, 490, 523, 524: AC changed to anomaly correlation

Line533, 535: PI changed to percentage increase

Line536: T/S changed to temperature and salinity

Line540, 655-658: PI changed to percentage increase

Line547: LUS changed to the Luzon Strait

Line562, 563, 564, 593: T/S changed to Temperature and salinity

Line566, 567, 568, 569, 576, 577, 578, 579, 580, 581, 584, 585, 587, 590, 591, 595, 596, 600, 601, 602, 603, 605, 606, 607, 611, 612, 613, 656, 657: **T** *changed to* temperature, **S** *changed to* salinity

Line571, 574, 583, 598, 650, 651: **3D** T/S *changed to* three-dimensional temperature and salinity

Line576, 578, 600, 602, 606: AC *changed to* anomaly correlation Line645: T/S *changed to* Argo

Third, the paper is not about improvements to SCSOFSv1, but rather about improvements "in" SCSOFSv2". The paper title as now indicates that you still only have v1, although it is improved. I think you would like to have a title as "The improvements in the .... (SCSOFS v2)"

Thanks. We have changed the paper's title to "The improvements in the regional South China Sea Operational Oceanography Forecasting System (SCSOFSv2)"

Fourth, v1.3 appears before it is clearly defined. Please modify the paper to avoid confusion. As most of the paper content for the comparison to v1 is not based on the assimilated system, there is no need to address the version 1.3. My understanding is that these improvements are present in final version 2. In this case, using v2 throughout the paper (except for the part evaluating subversions) would cause less confusion.

Thanks. We have changed the Figure2's title from SCSOFSv1.2 to SCSOFSv2 in line 192, and changed label from SCSOFSv1.3 to SCSOFSv2 in the Figure4.

Depending on the revision I will decide whether to send the manuscript to reviewers for a second round of review. Please provide reply to previous reviewers comments according to the new revisions.

Thanks for your good suggestions and reviewing for our manuscript again.

**Responding to Anonymous Referee #1**

**Summary:**

The manuscript represents updates of a South China Sea Operational Oceanography Forecasting System from version 1 to version2, which provide daily updated hydrodynamic forecasting in the South China Sea for the future 5 days. Comprehensive updates of model configuration and assimilation schemes have been mentioned. Among them, three major changes have been highlighted, namely the way of prescribing buoyancy flux, the tracer advection discrete scheme and data assimilation scheme. The model shows enhanced performance in the accuracy of the sea surface temperature and sea surface height.

Ocean states prediction has always been a challenging task and is of vital importance to the hazard prevention such as tropical cyclones and internal waves and so on. The South China Sea has unique monsoon system and topography and external Kuroshio forcing, which masks it a challenging region for ocean prediction. This paper is generally well written, and the way of improvement is well presented, which makes the comparison of results quite convincing. I believe this manuscript can be the basis of a useful publication after minor improvements.

The authors thank the reviewer for the insightful comments, and we completely agree with the questions and comments raised by the reviewer, which have helped us to improve the quality of the manuscript. We have carefully considered the reviewer's comments. Detailed replies to specific comments by the reviewer are presented below:

1 There are too many acronyms and I sometimes have to go back to look for its meaning. I would recommend keeping some acronyms frequently used by other articles, such as SCS and OFS and SST, but don't use abbreviation for only two words (such as RTOFS, PI) and abbreviation that is too hard to recall (such as RSUP3, U3H and C4V).

Line46: remove (GODAE);

Line52: remove (CONCEPTS);

- Line54: remove (RTOFS)
- Line56: remove (HYCOM)
- Line58: remove (JMA)
- Line61: remove (WNP)
- Line62: remove (CGOFS)
- Line76: remove (3DVAR)
- Line77: remove (4DVAR)

Line86: remove CMEMS, and ; Lellouche et al.

- Line172: remove (JMA)
- Line287: remove (RSUP3)

Line289: replace RSUP3 with the rotated split upstream third-order scheme

Line291: remove (hereafter referred to as U3H)

Line292: remove (hereafter referred to as C4V)

Line298: remove ,hereafter referred to as ISO

Line303: replace U3H with third-order upstream horizontal advection, replace C4V with fourth-order centered vertical advection, replace ISO with horizontal mixing on epi-neutral surfaces

**Line523, 524: replace GODAE with GOV**

2 Line 1: The improvements to the regional South China Sea Operational Oceanography Forecasting System include all the previous versions. I suggest removing the acronym of "(SCSOFSv1)".

Thanks for your pointing out this. But the acronym of "(SCSOFSv1)" and its version number are added following with the editor's review before public discussion. And now we have changed it to SCSOFSv2.

3 Line 199: Please clarify the wards in this headline. What are the highlights and sensitive updates respectively?

We think the three improvements mentioned in this section are most important than others mentioned in section 2. They have significantly improved the model skill of SCSOFS from different aspects. And we have added some explanation it in Line 212.

4 Section 3.1, Line 220: The use of bulk formulation to calculate the buoyancy fluxes is reasonable, but it is not a real negative feedback because the atmospheric forcing, such as air temperature, relative humidity are prescribed, which are not adjusted from the modelled SST. Please clarify this sentence, e.g. how the SST is improved through the use of bulk formulation should be further elaborated.

Thanks for pointing out this. We agree that the use of bulk formulation does not represent a real negative feedback like ocean-atmosphere coupled model, since the atmospheric forcing is prescribed. But it still can play a role with negative feedback function to the simulation of SST, because the calculation of sensible heat flux, latent heat flux and longwave radiation uses SST calculated by ocean model. We have added a reference Li et al. (2021) to elaborate the calculation of three air-sea fluxes in Line 230 and Line 275.

5 Figure 3. For the SCSOFSv1, the area north of 24°N in the BulkFormula is even warmer than that in the no BulkFormula experiment. But the SST is much improved in the SCSOFSv2. Later results in Section 3.2 indicate that it may be related to the improved advection and mixing scheme. Please further explained this by providing more information.

Thanks. We think this should be considered as two different problems separately. For the first one, the area north of 24°N in the BulkFormula is even warmer than that in without BulkFormula experiment, it should be attributed to the local complicated airsea interactions in the area and tidal mixing is missing in the model. For the second one, the SST is much improved in the SCSOFSv2, it mainly due to the improved advection and mixing scheme representing the vertical heat transport well, then the surface layer affected by subsurface and deep layer processes. We have added more explanations in Line 275, Line 279, and Line367-Line369.

6 Section 3.2: What about the temperature and salinity bias in the subsurface layer in the AAG scheme combination?

Please refer to Figure 13 and 14 for the results from v1.3. We have modified it in Line 367.

7 Line 335. Please explain the improvement of temperature and salinity with more detail. What is the diffusion term and advection term look like in the AGG scheme

**combination?**

Thanks. The harmonic mixing scheme is used for both viscosity for momentum and diffusion for tracers in horizontal. And Mellor-Yamada Level-2.5 vertical mixing closure scheme is used for both momentum and tracers. This has been explained in Line 194-Line199.

8 Line 360: Why do you set observational error for the SLA and SST as 0.09 cm and 0.5 °C? The along-track satellite data contains high-frequency noise, especially for the shallow area (Zhou et al., 2015). How do you filter out this noise?

Zhou, Xiao-Hui, Dong-Ping Wang, and Dake Chen. "Global wavenumber spectrum with corrections for altimeter high-frequency noise." Journal of Physical Oceanography 45.2 (2015): 495-503.

Thanks for pointing out this. We have revised this in text Line 381-Line 387.

We have used filtered SLA for assimilation products specially from AVISO, which is filtered with 20-day cutoff-period but not subsampled unlike other L3 products.

9 Line 428: In the section 4, I suggest introducing why the SST, SLA and T/S profiles are used to validate the model. Are those element enough to represent the outputs from the model? In addition, you provide the importance of SST in Line 210 to connect with the sea surface atmospheric forcings, but what kind of validations is related to section 3.2 and 3.3.

Thanks. We employ the IV-TT Class4 verification framework to validate the model, which is an international common verification metric. The main reason for using the SST, SLA, and T/S profiles is that public data are easy to obtain and has accumulated plenty of data. Of course, these elements are not enough to represent the outputs from model. But we can not get enough currents observation data to validate our model. Validations with the subsurface layer temperature and salinity using T/S profiles mainly relate to section3.2, all those validations are related to section 3.3, since all those three kinds of data have been assimilated by MOOAS.

10 Table 1: In Line 145 you mentioned the the new SODA 3.3.1 and 3.3.2 reanalysis were used, but in Table 1 you still mention the SODA 2.2.4, please check all the settings in this table.

Thanks. We have added one more information in new Table 2, as "Changing the **open boundary data** from SODA 2.2.4 monthly mean to SODA 3.3.1 and 3.3.2" while upgrading from v1.2 to v1.3.

11 Figure 13: Why the RMSE of temperature is suddenly large in June in Fig. 13b? In the paper of a recently published paper (Ding et al., 2021), there is also a similar large bias in June, can you provide some explanations to this?

Ding, R., Xuan, J., Zhang, T., Zhou, L., Zhou, F., Meng, Q., and Kang, I. (2021), Eddy-Induced Heat Transport in the South China Sea, Journal of Physical Oceanography, 51(7), 2329-2349, doi: 10.1175/JPO-D-20-0206.1.

Thanks for pointing out this issue. We have checked the original data in detail, and found that there were two Argo temperature profiles with bad quality as shown in following figures. The temperature observation is almost 40 °C at surface, and more

than 10°C below 1000m layer. We have removed the two profiles, recalculated RMSE and replotted the figure 13b. We are sorry about this.

12 Please check the font size in all the figures to make sure it is clear. Also add title for similar plots.

Thanks. We have replotted the figure 2, 3, and 12 with larger font size.

**Responding to Anonymous Referee #2**

The authors describe the evolution of the South China Sea operational oceanography forecasting system (SCSOFS), which significantly improve the quality of the system. The manuscript provides a detailed description of different versions of the ROMS model which were used for SCSOFSv1 and SCSOFSv2 and compare the 11-year long free run of both models. The results show various improvements for the changes that have been made to the operational system. It provides useful suggestions for the community. My opinion is that the paper is worthy of publication in GMD after minor modifications. But several specific aspects of the paper detail and analysis should be considered first.

Thanks for your confirmation to our manuscript.

**2 Specific comments**

The grammar and quality of the writing needs to be improved before publication: many sentences are over-complicated or confusing (Example lines: 113-114, 170-174, 341-343....). It would benefit from a thorough revision by someone with proficient written English skills.

We have invited a professional of similar academic background with proficient English written skills to proofread and revise our manuscript thoroughly according to the reviewer's comments.

Line116: "sea areas" change to "areas"

Line359-362: "The results of SCSOFSv2 show much more areas with lower SST than OSTIA in the central Pacific Ocean, comparing to the results of SCSOFSv1 and BulkFormula, which can be attributed to the new scheme combination." change to "Comparing with the results of SCSOFSv1 and BulkFormula, less SST hot bias versus

OSTIA is found in the central Pacific Ocean for the results of SCSOFSv2, which can be attributed to the new scheme combination."

Line 99: the section title is not clear, which datasets? Not all datasets used in the article are described here.

Thanks. We have changed to "input datasets" in Line102.

Line 142: For what reasons are the initial conditions not deduced from soda 3.3 in correspondence with the boundary conditions?

Thanks. One reason is that the resolution of GDEMV3  $(0.25^{\circ} \times 0.25^{\circ})$  is finer than SODA3.3  $(0.5^{\circ} \times 0.5^{\circ})$ , and GDEMV3 is derived from plenty of observations with good accuracy. The other reason is that our model was started from climatological run with climatological initial, boundary, and surface forcing conditions. We just selected climatological initial conditions with finer resolution.

Lines 180-184: Why does the inclusion of Guam at the open boundary have such a significant influence on the latitude of the NEC in the model? This need be explored or explained.

Thanks. It is because that the Guam Island changes the bathymetry from over 3500m to below 500m suddenly, and as a big block to the NEC once flowing into the model domain from eastern lateral boundary. We have explained it in Line 179.

Lines 189-193: Why these choices as opposed to the ones used in v1? What is special about 3rd order and biharmonic?

Thanks for pointing out this. I have clarified it between Line194 and Line199, and added one more table1. Actually, the third-order upstream and fourth-order centered schemes are used for momentum in SCSOFSv1, which are default settings in ROMS. We do not use biharmonic mixing scheme, just harmonic mixing for both SCSOFSv1 and SCSOFSv2.

Line 227: Any reference for OSTIA first appeared in the text? Is it provided to SCSOFS directly by the Metoffice or by other means?

In our manuscript, we have already cited the paper of Donlon et al.,2012 in the end of this sentence. OSTIA data is open access to public. We have already change 'provide' to 'produced' in Line 242 to avoid possible misunderstanding.

Line 326: the sentence on the conclusion may come after the lines 327-337.

Thanks, we have deleted the conclusion in Line341

Line 355: Where do you get the data from in SCSOFSv2, is it still the same servers? Where do the ARGO observations come from?

Thanks. Yes, we have got all SLA, OSTIA, ARGO observation data from the website https://marine.copernicus.eu of CMEMS in SCSOFSv2, it is the same with SCSOFSv1.

Line 373: "around" is not an exact annotation, need to point out exactly.

Thanks. We have changed it to ", with 30 days before and after" in Line 402.

Line 383: "for the calculation of innovations" change to "when calculating innovations" Thanks. We have done it in Line 412.

Line 385: Lee et al., 22-25 Jun, format error!

Thanks. We have removed "22-25 June" in Line 416.

Line 619: What are the known biases of the system? Some are described in the article; can you summarize them in the conclusion? Can you elaborate the summary in terms of what is the added value of this regional system compared to existing global systems? Thanks for your good suggestions. We have added it in conclusions section. For the comparison to existing global system, it is not done in this article, so we can not give more information. We will try to do that with many existing global systems in later. Section 2: A table summarizing the v2 and v1 features would be useful. Thanks for your good suggestions. We have added a new table1.

Fig. 8: Why 7-day FGAT? Why not 3-days or 10-days? Can you explain more on this? The length of First Guess at Appropriate Time (FGAT) is associated with the data assimilation window, which is taken as 7 days in our data assimilation system. The length of assimilation window is decided by several factors. 1) The observations of Sea Level Anomaly (SLA) and temperature/salinity profiles are relatively sparse, therefore, the window is enlarged enough to include more observations. 2) the true state evolves with time, therefore, the window is shortened, as the difference between observation and background increases with time. 3) The data assimilation system runs every 7 days, and forecasting system provides the forecast of 7 days in future, therefore, 7 days of window is adopted to collect the more observations, and also to avoid reusing some observations. The window of 7 days is also used by other operational oceanography system, such as the PSY4 in Mercator. Therefore, the length of 7 days is appropriate for the assimilation window, and also for the FGAT.

---

## Author Response (AR3)

**Comment on gmd-2021-235**

**Responding to Topical Editor**

Dear authors,

There are still grammar issues in many places. Please work on the paper before resubmission.

Best wishes

Qiang Wang

We have asked for a professional service from LerPub company to proofread and revise our manuscript thoroughly according to the editor's comments. Please see the certificate of English language editing as follow.

**Certificate of English Language Editing**

[Figure]

**Manuscript Title:**

Improvements to the regional South China Sea Operational Oceanography Forecasting System (SCSOFSv2)

**Date of Revision:**

December 20, 2021

Abstract:

The South China Sea Operational Oceanography Forecasting System (SCSOFS), constructed and operated by the National Marine Environmental Forecasting Centre of China, has been providing daily updated hydrodynamic forecasting in the SCS for the next five days since 2013. This paper presents recent comprehensive updates to the configurations of the physical model and data assimilation scheme in order to improve the forecasting skill of the SCSOFS. This paper highlights three of the most sensitive updates, including the sea surface atmospheric forcing method, the discrete tracer advection scheme, and modification of the data assimilation scheme. Inter-comparison and accuracy assessment among the five versions were performed during the entire upgrading process using the OceanPredict Inter-comparison and Validation Task Team Class4 metrics. The results indicate that remarkable improvements have been made to the SCSOFSv2 with respect to the original version known as SCSOFSv1...

This document certifies that the manuscript listed above was copy edited for English language by LetPub, with regard to grammar, punctuation, spelling, and clarity. All of our language editors are native English speakers with long-term experience in editing scientific and technical manuscripts. We are committed to leveling the playing field for researchers whose native language is not English.

- Documents receiving this certification should be regarded as having undergone professional editorial revision for English language before submission. However, the authors may accept or reject LetPub's suggestions and changes at their own discretion and LetPub does not have editorial control over the submitted documents.
- The language quality of the submitted document is the sole responsibility of the submitting authors subject to those authors' adherence to LetPub's revisions and instruction. LetPub's provision of service does not constitute a guarantee or endorsement of the authors' work herein.
- Neither the research content nor the authors' intended meaning were altered in any way during the editing process.
- If you have any questions or concerns about this edited document, please contact us at support@letpub.com

[Figure]

LetPub is an author service brand owned and operated by Accdon LLC. Headquartered in the Boston area, we are a full-spectrum author services company with a large team of US-based certified language and scientific editors, ISO 17001 accredited translators, and professional scientific illustrators and animators. We advocate ethical publication practices and are an official member of the Committee on Publication Ethics (COPE).

For more information about our company, services, and partnership programs, please visit www.letpub.com.

© 2021 Accdon, LLC. All Rights Reserved.  Tel: 1-781-202-9968  Email: info@accdon.com  Address: 400 Fifth Ave, Suite 530, Waltham, MA 02451, United States

---

## Author Response (AR4)

Comment on gmd-2021-235

**Responding to Topical Editor**

The manuscript will be accepted for publication after a few very minor technical corrections.

Many thanks for your confirmation to our manuscript.

1) In my last comments I forgot to mention that I agree with one of the reviewers that the use of "negative feedback" is not correct. It is simply an "implicit SST restoring effect", and has nothing to do with physical feedback mechanism. Please correct them ("negative feedback" appears at 5 places).

Thanks for pointing out this for us. We have modified them as follow:

Line 236: "an effective negative feedback mechanism can form between the SST and the SST-related heat fluxes." changed to "an implicit SST restoring effect can be formed between the SST and the SST-related air-sea heat fluxes."

Line260: "introducing the effective negative feedback mechanism between the model's SST and the air-sea heat flux using the COARE 3.0 bulk algorithm," changed to "introducing the implicit SST restoring effect using the COARE 3.0 bulk algorithm,"

Line275: "due to the effective negative feedback mechanism" Changed to "due to the implicit SST restoring effect"

Line279: "with an effective negative feedback mechanism" changed to "with an implicit SST restoring effect"

Line635: "an effective negative feedback mechanism" changed to "an implicit SST restoring effect"

2) The word "feedbacked" on page 27 should be "feedback"

We have done.

Line533: replace feedbacked with feedback

3) in the abstract, "the five versions" --> "five sub-versions"

We have done.

Line18: replace versions with sub-versions

4) page 32, 647-648, can well ... preserve --> can well preserve ...

We have done.

Line647: replace well the characteristics with well preserve the characteristics

Line648: remove preserve